METHODS AND RESOURCES

# IntAct: A nondisruptive internal tagging strategy to study the organization and function of actin isoforms

**Maxime C. van Zwam**[1], **Anubhav Dhar**[2], **Willem Bosman**[1], **Wendy van Straaten**[1], **Suzanne Weijers**[1], **Emiel Seta**[1], **Ben Joosten**[1], **Jeffrey van Haren**[3], **Saravanan Palani**[2]*, **Koen van den Dries** [1]*

1 Department of Medical BioSciences, Radboud University Medical Center, Nijmegen, the Netherlands,
2 Department of Biochemistry, Division of Biological Sciences, Indian Institute of Science, Bangalore, India,
3 Department of Cell Biology, Erasmus MC, Rotterdam, the Netherlands

* spalani@iisc.ac.in (SP); koen.vandendries@radboudumc.nl (KvdD)

## Abstract

Mammals have 6 highly conserved actin isoforms with nonredundant biological functions. The molecular basis of isoform specificity, however, remains elusive due to a lack of tools. Here, we describe the development of IntAct, an internal tagging strategy to study actin isoforms in fixed and living cells. We identified a residue pair in β-actin that permits tag integration and used knock-in cell lines to demonstrate that IntAct β-actin expression and filament incorporation is indistinguishable from wild type. Furthermore, IntAct β-actin remains associated with common actin-binding proteins (ABPs) and can be targeted in living cells. We demonstrate the usability of IntAct for actin isoform investigations by showing that actin isoform-specific distribution is maintained in human cells. Lastly, we observed a variant-dependent incorporation of tagged actin variants into yeast actin patches, cables, and cytokinetic rings demonstrating cross species applicability. Together, our data indicate that IntAct is a versatile tool to study actin isoform localization, dynamics, and molecular interactions.

## Introduction

Actin plays a central role during fundamental biological processes including cell division, shape maintenance, motility, and contractility. In birds and mammals, actin has 6 isoforms, also called isoactins, which are encoded by different genes and expressed in a tissue and time-specific manner during development, homeostasis, and pathology [1–3]. All 6 isoactins have nonredundant functions as indicated by the discovery of disease-causing mutations in each of the genes encoding the isoforms [4]. Although exceptions exist, it is generally acknowledged that 4 isoactins are expressed in muscle cells and 2 are ubiquitously expressed across tissues. The 2 ubiquitous isoactins, nonmuscle β- and γ-actin, display the highest similarity with only 4 different residues at their N-terminus [5]. Despite this similarity, nonmuscle β- and γ-actin play specific roles in many actin-controlled processes including cell–cell junction formation [6], axon development [7], cell division [8,9], and cell migration [10,11]. While it has been

**Data Availability Statement:** Numerical data is available in S1 Data. The raw, uncropped western blots are available in S1 Raw Images. The flow

cytometry data is publicly available through the FlowRepository, accession number: FR-FCM-Z776.

**Funding:** This work was financially supported by a Department of Biotechnology-Wellcome Trust India Alliance intermediate fellowship (IA/I/21/1/505633), SERB SRG grant (SRG/2021/001600) and an Indian Institute of Science (IISc) start-up grant awarded to S.P., support from Convergence Flagship CIFIC (Convergence Program Erasmus MC, EUR, TU Delft) to J.v.H., as well as intramural funding of the Radboudumc and an NWO KLEIN grant (OCENW.KLEIN.494) awarded to K.v.d.D. The funders had no role in study design, data collection and analysis, decision to publish, or preparation of the manuscript.

**Competing interests:** The authors have declared that no competing interests exist.

**Abbreviations:** ABP, actin-binding protein; BSA, bovine serum albumin; HDR, homology-directed repair; LP, long pass; MBS, mass beam splitter; PCC, Pearson correlation coefficient; PMT, photomultiplier tube; ROI, region of interest.

demonstrated that isoactin-specific posttranslational modification [12,13], nucleation [8,9], and translation speed [13,14] contribute to the nonredundant role of β- and γ-actin in cellular processes, many questions on the molecular principles that govern the differential function of isoactins remain unanswered, including isoform-specific dynamics or the spatiotemporal control of isoform-specific association with actin-binding proteins (ABPs). This is primarily due to the limited possibilities to specifically probe actin isoforms for biochemical and cell biological assays.

Common tools to label the actin cytoskeleton such as phalloidin [15] in fixed cells or Lifeact [16], F-tractin [17], UtrophinCH [18], and anti-actin nanobodies [19–21] in living cells do not discriminate between isoactins. Specific antibodies for nonmuscle β- and γ-actin have been previously developed against the distinct N-terminus of isoactins [11] yet the N-terminal epitope is only available under specific permeabilization-fixation conditions and methanol treatment is required for optimal epitope accessibility [11]. This prevents many downstream applications including co-immunoprecipitation assays or targeting in living cells. C-terminal fusions of actin cannot be used to study isoactin biology since they only poorly assemble into filaments [22,23]. N-terminal fusions of actin have been used to study isoactin differences in cells [14,24,25], but the reporter tags are known to significantly interfere with actin dynamics [16], nucleation [26], and molecular interactions [16,26–28]. Moreover, N-terminal fusion prevents isoactin-specific and nonspecific posttranslational modifications crucial for proper actin function such as arginylation [12,13] and acetylation [29,30]. Attempts to tag yeast actin internally with a tetra-cysteine tag for contractile ring visualization were unsuccessful since the modified actin subunits were rejected by formins during filament elongation [31]. An extended search for other internal sites that may be permissive for epitope tagging of actin has not been performed so far.

Here, we describe the development of an internal tagging strategy to study isoactin organization and function, which we call IntAct. For this, we first performed a microscopy-based screen for 11 internal actin positions and identified 1 residue pair that allows epitope tagging of actin based on its assembly into various actin-based structures including lamellipodia, filopodia, and stress fibers. To prove its versatility and usability, we engineered CRISPR/Cas9-mediated knock-in cell lines with various antibody- and nanobody-based epitope tags in the identified position and demonstrate that the internally tagged actins are properly expressed and that the integration into filaments is unperturbed. By performing immunofluorescence, pulldown experiments, and live-cell imaging with the internally tagged actins, we show that IntAct can be used to study isoactin localization, molecular interactions, and dynamics. Lastly, we show that internally tagged nonmuscle β- and γ- actin mimic the differential distribution of wild-type isoforms and demonstrate the incorporation of tagged actin variants into yeast actin patches, cables, and cytokinetic rings indicating the possibility to extend the use of IntAct to other species. Altogether, our results indicate that IntAct can provide unique insights into the isoactin-specific molecular principles that regulate cellular processes such as division, motility, and contractility.

## Results

### T229/A230 actin residue pair is permissive for epitope tag insertion

To identify a permissive residue pair to internally tag the actin protein, we first performed a medium-scale screen and tagged β-actin at 11 distinct residue pairs with a FLAG tag (**Fig 1A**). These residue pairs were carefully selected, ensuring that at least one of the residues is part of an unstructured region [32] and that both residues do not regulate F-actin interactions [33]. Furthermore, the first 40 residues were avoided since the coding mRNA for this region is

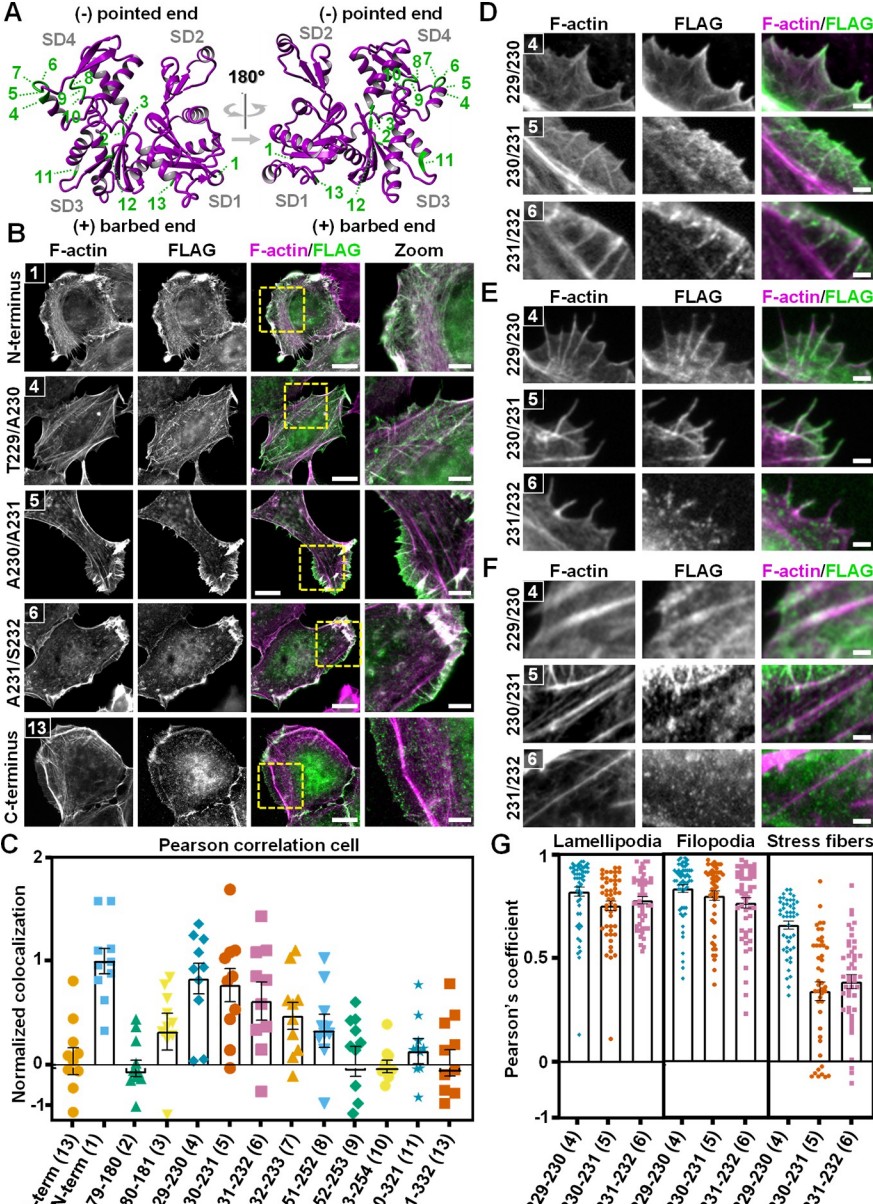

**Fig 1. Identification of actin T229/A230 as a permissive target site for epitope tag integration.** (**A**) Crystal structure of uncomplexed globular actin (magenta ribbon, PBD accession number: 1J6Z [32]) indicating the 11 internal target positions (green) as well as the N- and the C-terminus (1 and 13, respectively). (**B**) Representative widefield immunofluorescence images of F-actin (magenta) and FLAG (green) in HT1080 cells that overexpress the tagged β-actin variants. Shown are 3 internally tagged variants and the N- and C-terminally tagged β-actin. The remaining 8 internally tagged variants are shown in **S2 Fig**. Scale bar: 15 μm. Scale bar zoom: 5 μm. (**C**) Colocalization analysis of the microscopy results in **B** showing the normalized Pearson's coefficient for each of the actin variants. Individual data points indicate single cells and in total, at least 10 cells from 2 independent experiments were included in the analysis. Bars represent the mean value and error bars represent SEM. (**D–F**) Representative images of (**D**) lamellipodia, (**E**) filopodia, and (**F**) stress fibers. Cells that overexpress the tagged β-actin variants were stained for F-actin (magenta) and FLAG (green). Scale bar: 2 μm. (**G**) Pearson's colocalization analysis for the images in **D–F**. Individual data points indicate single lamellipodia, filopodia, or stress fibers and in total, at least 40 structures in at least 20 different cells from 2 independent experiments were included in the analysis. Bars represent the mean value and error bars represent SEM. The numerical data underlying this figure can be found in **S1 Data**.

involved in the different translation kinetics of actin isoforms [13]. Eventually, 2 of the 11 selected positions were located in subdomain 3, 7 in subdomain 4, and 2 were close to the ATP binding pocket (**S1 Fig**). C- and N-terminally tagged β-actin were included in the screen as a negative and positive control for filament integration, respectively. We chose the FLAG tag as an epitope for our screen because of its frequent use, small size (8 amino acids, DYKDDDDK), and availability of a highly specific and well-characterized antibody [34].

To evaluate the integration of the tagged actins within the actin cytoskeleton, we overexpressed the 13 actin variants in human fibrosarcoma cells (HT1080; **Figs 1B**, **1C and S2**) and retinal pigment epithelium cells (RPE1, **S3 Fig**) and performed an immunofluorescence staining for the FLAG tag and phalloidin as a total F-actin marker (**Figs 1B**, **S2 and S3A**). Interestingly, by visual inspection, we observed that most of the internally tagged actins were diffusely present within the cytosol with 3 notable exceptions (T229/A230, A230/A231, and A231/S232). Of these 3 variants, A230/A231 and A231/S232 only seemed to present a clear overlap with actin at the cell periphery but the T229/A230 overlapped almost entirely with the actin signal, similarly to N-terminally tagged actin. To quantify these observations, we first performed a Pearson correlation coefficient (PCC) analysis on entire cells (**Figs 1C** and **S3B**). As expected, N-terminally tagged actin showed a high degree of colocalization, while C-terminally tagged actin showed almost no colocalization (**Figs 1C** and **S3B**). We therefore performed a unity-based normalization, adjusting the PCC of the C- and N-terminus to zero and one, respectively, and normalized the other values within this window. While most of the internally tagged actin variants showed little to no colocalization in both cell types, the T229/A230 variant consistently displayed a high PCC in both HT1080 and RPE1 cells (raw PCC = 0.68 and 0.75, respectively). Interestingly, for the A230/A231 and A231/S232 variant, we observed a very low colocalization in RPE1 cells, but a relatively high colocalization in HT1080 cells. We therefore sought to further dissect the integration of these 3 variants into the actin cytoskeleton of HT1080 cells and quantified the colocalization within individual actin-based structures (**Fig 1D–1F**). While we see a high degree of colocalization for all 3 variants in lamellipodia and filopodia, only the T229/A230 variant shows a high degree of colocalization also in stress fibers (**Fig 1G**), demonstrating the unique ability of the T229/A230 variant to be integrated in each of these common actin-based structures. Together, these results strongly suggest that the T229/A230 residue pair in actin is permissive for epitope tag insertion.

## T229/A230 epitope tag insertion does not impair actin expression or assembly into filaments

We next applied CRISPR/Cas9-mediated homology-directed repair (HDR) to genetically introduce various tags into the identified internal position at the genomic locus of β-actin. To demonstrate the versatility of the T229/A230 position as well as its usability for various downstream cell biological and biochemical applications, we included both antibody-based epitope tags (FLAG (DYKDDDDK), AU1 (DTYRYI), and AU5 (TDFYLK)) and the nanobody(Nb)-based (ALFA tag (PSRLEEELRRRLTEP)) [35]. Flow cytometry data shows that we retrieved a similar degree of knock-in efficiency for all the different tags (**S4A Fig**). To specifically select for HDR events, we performed a ouabain-based co-selection procedure [36] and again retrieved similar degree of knock-in efficiency for all the tags (**S4B Fig**). For AU5, FLAG, and ALFA, we continued to select clonal cell lines and for all 3 tags, based on western blot and sequencing analysis, we were able to retrieve clonal cell lines that exclusively produce internally tagged β-actin. For AU5 and FLAG, the clones were hemizygous and for ALFA, we obtained a homozygous knock-in clone (**S4C and S4D Fig**). To also demonstrate that it is possible to tag multiple isoforms in the same cell, we performed a knock-in of the FLAG tag in γ-actin in the

already established homozygous ALFA-β-actin HT1080 cells. Immunofluorescence labeling of FLAG and ALFA in these cells confirmed the possibility to simultaneously tag β- and γ-actin at the T229/A230 position with different epitope tags (**S5A Fig**).

Previously published results indicated that a heterozygous knock-in of GFP into the genomic locus of β-actin for N-terminal tagging leads to a dramatic decrease of protein expression from the modified allele [27]. After generation of our knock-in cell lines, we therefore first determined if internally tagging actin at position T229/A230 leads to an altered actin expression in the clonal cell lines. For this, we first used the heterozygous and hemizygous FLAG-β-actin cells since FLAG causes a gel shift on western blot, allowing a direct comparison between tagged and wild-type actin (**Figs 2A and S5B**). Quantification of the western blots demonstrated that the amount of FLAG-β-actin was similar to wild type in heterozygous FLAG-β-actin cells (**Fig 2B**) and that the expression of both β-actin and γ-actin did not change in hemizygous FLAG-β-actin cells (**S5C Fig**). We next evaluated actin protein expression in the homozygous ALFA-β-actin cells and this showed that β-actin was approximately 25 to 30 percent lower in the ALFA-β-actin cells compared to parental HT1080 cells (**Fig 2C** and **2D**). Although we suspect that this decrease is attributed to clonal variation, we wanted to exclude compromised global actin regulation in the ALFA-β-actin cells. We therefore evaluated the expression of γ-actin and α-smooth muscle actin (α-SMA) since a genetic loss of β-actin has been shown to induce the expression of these isoforms [37]. Importantly, we only observed a very minor increase (approximately 8 percent) in γ-actin expression and no induction of α-SMA in the ALFA-β-actin cells (**S5D–S5F Fig**), strongly suggesting that global regulation of actin isoform expression is not perturbed by genetically tagging β-actin. Next, since we deliberately chose an internal position over the N-terminus, we also investigated if β-actin remains posttranslationally modified by arginylation. Here, we found that β-actin is still arginylated in both the FLAG-β-actin as as well as the ALFA-β-actin cells (**S6A Fig**). Finally, we aimed to establish if tagged actin translocates to the nucleus since it is increasingly recognized that actin plays an important role in regulating nuclear processes including gene transcription and chromosome organization [38]. By performing subcellular fractionation experiments, we observed ALFA-β-actin in both the nuclear as well as the cytosolic fraction, indicating that ALFA-β-actin translocates to the nucleus (**S6B Fig**).

Next, we assessed the incorporation of the tagged actins into the cytoskeleton by performing immunofluorescence labeling followed by widefield microscopy (**Fig 2E**). Pearson colocalization analysis demonstrated that all the knock-in tagged actins have a strong overlap with F-actin, indicating that they are well incorporated in actin filaments (**Fig 2F**). Moreover, high-resolution images of actin and nonmuscle myosin IIA in the homozygous ALFA-β-actin cells indicated that stress fibers have a similar architecture as compared to parental cells (**S7 Fig**). Since the ALFA tag allows intracellular detection of the tagged actins in living cells, we also performed live-cell imaging with the ALFA-β-actin cells. For this, we overexpressed ALFA-Nb-mScarlet in the ALFA-β-actin cells and evaluated its colocalization with F-actin by co-transfecting Lifeact-GFP and determining the Pearson's coefficient at multiple time points (**S8 Fig** and **S1 Movie**). This demonstrated that, also in living cells, there is a very high correlation (PCC = >0.8) between the fluorescence intensity of the tagged actins and total actin (**S8B Fig**). Of note, expression of GFP and ALFA-Nb-GFP/mScarlet in parental cells or GFP in ALFA-β-actin cells resulted in a diffuse cytosolic fluorescence signal indicating that the observed colocalization is specific (**S9 Fig**). Together, these findings further support the notion that the T229/A230 internally tagged β-actin is properly assembled into actin filaments.

To corroborate our microscopy results, we sought to biochemically determine the F/G-actin ratio of tagged and wild-type actin, for which we used the heterozygous FLAG-β-actin and homozygous ALFA-β-actin cells. The results from these experiments demonstrated that

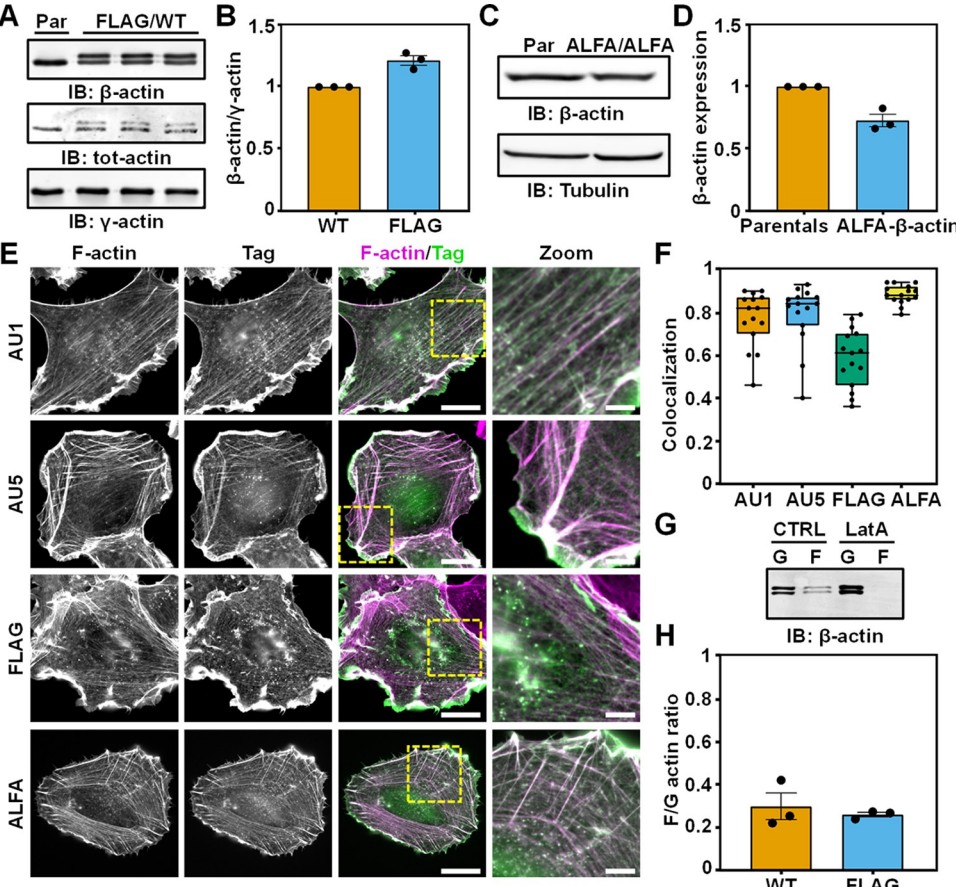

**Fig 2. Actin functionality unperturbed by T229/A230 epitope integration.** (**A**) Western blot of β-actin, total actin and γ-actin in parental HT1080 (Par) and 3 independent heterozygous FLAG-β-actin HT1080 clones (FLAG/WT). (**B**) Quantification of β-actin protein expressed by the WT allele and the FLAG allele as shown in **A** and normalized to γ-actin. Individual data points represent 3 different heterozygous FLAG-β-actin HT1080 clones. Bars represent the mean value and error bars represent SEM. (**C**) Representative western blot showing β-actin expression in parental HT1080 (Par) and homozygous ALFA-β-actin HT1080 cells (ALFA/ALFA). (**D**) Quantification of the β-actin expression from the western blot shown in **C** and normalized to tubulin. Individual data points represent 3 independent western blots. Bars represent the mean value and error bars represent SEM. (**E**) Representative widefield images from cells that have a CRISPR/Cas9-mediated knock-in of AU1, AU5-t, FLAG, or ALFA tag in β-actin. Cells were labeled for F-actin and an antibody/nanobody against the respective tag to visualize F-actin (magenta) and the tagged β-actin (green). Scale bar: 15 μm. Scale bar zoom: 5 μm. (**F**) Colocalization analysis of the microscopy results in **E** showing the Pearson's coefficient for each of the internally tagged actins. Individual data points indicate single cells and in total 15 cells from 2 independent experiments were included in the analysis. Box plots indicate median (middle line), 25th, 75th percentile (box) and minimum and maximum (whiskers). (**G**) Representative western blot of G-actin and F-actin fraction in heterozygous FLAG-β-actin HT1080 cells that were left untreated or treated with latrunculin A. (**H**) Quantification of the F/G-actin ratio for β-actin expressed by the WT allele and FLAG allele from the western blot shown in **G**. Individual data points represent 3 independent western blots. Bars represent the mean value and error bars represent SEM. The numerical data underlying this figure can be found in **S1 Data**.

the F/G-actin ratio for FLAG-β-actin and ALFA-β-actin was indistinguishable from wild-type actin (**Figs 2G, 2H** and **S10**), indicating that the internally tagged β-actin variants were normally integrated into actin filaments. Moreover, expression of the ALFA-Nb-GFP did not alter the F/G-actin ratio in the ALFA-β-actin cells, demonstrating that the interaction of the nanobody with the tagged actin does not influence its integration (**S10 Fig**).

Together, these results in fixed and living cells with multiple epitope tags suggest that the T229/A230 residue pair in actin is a versatile position for epitope tagging with only a minor

impact on actin expression and no measurable effect on the ability of actin to integrate into filaments.

## Internally tagged actins remain associated with established G-actin and F-actin interactors

To study the molecular interactions of internally tagged β-actin, we performed a co-immuno-precipitation assay and western blot analysis using FLAG-β-actin and ALFA-β-actin. Since co-immunoprecipitation of actin only allows the investigation of monomeric G-actin interactors, we first evaluated the association of the well-established G-actin binding proteins cofilin and profilin. From the results, we first concluded that both FLAG- and ALFA-β-actin could be immunoprecipitated from the lysates (Fig 3A–3C), demonstrating the availability of the epi-tope tags in whole cell lysates. More importantly, we also observed that cofilin and profilin co-immunoprecipitate with both FLAG-β-actin (Fig 3A and 3B) and ALFA-β-actin (Fig 3C) strongly suggesting that the internally tagged actins maintain their ability to associate with these important G-actin regulators. For ALFA-β-actin, we further investigated its ability to associate with the actin nucleating formins DIAPH1 and FMNL2, which we also expect to be co-immunoprecipitated with actin under these conditions. The results from these experiments show that both DIAPH1 and FMNL2 associate with ALFA-β-actin (Fig 3D), strongly suggest-ing that the internal tag does not prevent the association of actin with formin family members, either directly or indirectly through profilin. Lastly, we also determined the association of ALFA-β-actin with the well-known F-actin crosslinkers myosin IIA and α-actinin. Using F-actin co-sedimentation of whole cell lysates, we observed similar levels of myosin IIA and α-actinin in the F-actin fraction (S11 Fig). These findings are in alignment with our myosin IIA immunofluorescence results and strongly suggest that these major F-actin binding proteins still properly associate with filaments that contain internally tagged actins.

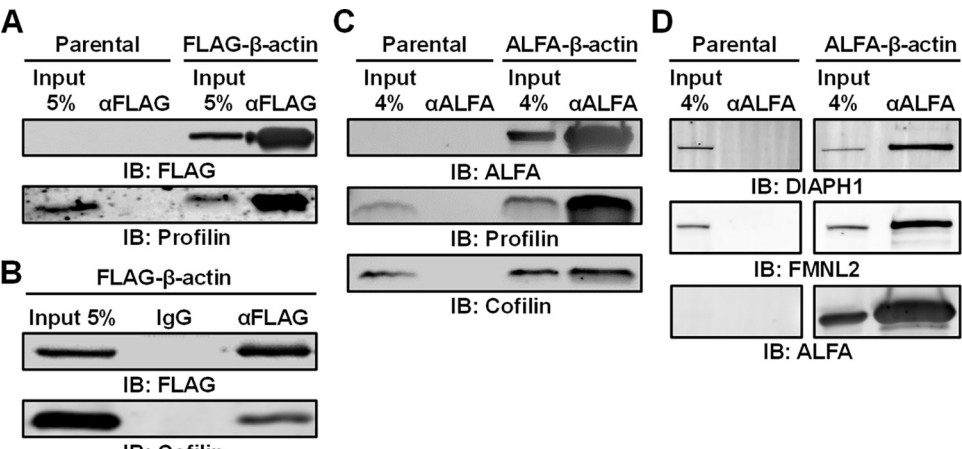

**Fig 3. FLAG- and ALFA-β-actin interact with profilin, cofilin, and formin family members DIAPH1 and FMNL2.** (**A**) Representative western blot showing the co-immunoprecipitation of FLAG-β-actin and profilin using an anti-FLAG antibody in the FLAG-β-actin HT1080 cells. Co-immunoprecipitation performed on parental HT1080 was included as a control. (**B**) Representative western blot showing the co-immunoprecipitation of FLAG-β-actin and cofilin using an anti-FLAG antibody. IgG was included as a negative control. (**C**) Representative western blot showing the co-immunoprecipitation of ALFA-β-actin and profilin and cofilin using an anti-ALFA nanobody in the ALFA-β-actin HT1080 cells. Co-immunoprecipitation performed on parental HT1080 was included as a control. (**D**) Representative western blot showing the co-immunoprecipitation of ALFA-β-actin, mDia1, and FMNL2 using an anti-ALFA nanobody in the ALFA-β-actin HT1080 cells. Co-immunoprecipitation performed on parental HT1080 was included as a control.

## Actin retrograde flow and cell proliferation and migration are not affected by β-actin internal tagging

Fluorescent fusions of actin are known to affect actin retrograde flow at the cell front, likely due to the large fluorescent reporter tag [16]. To evaluate whether actin retrograde flow is unperturbed by introducing the internal tag at position T229/A230, we determined the actin flow at lamellipodia using live-cell imaging (**Fig 4A**). For this, we transfected ALFA-β-actin cells with Lifeact-GFP or ALFA-Nb-GFP and performed time lapse imaging with Airyscan super-resolution microscopy. Parental cells transfected with Lifeact-GFP were included as a control since the expression of Lifeact has been demonstrated to not affect the actin retrograde flow at the cell front [16]. Importantly, we observed actin flow at lamellipodia in all of the conditions indicating no gross defects in the formation of these structures by the internal ALFA tag (**S2–S4 Movies**). Moreover, by quantitative analysis of the kymographs from the time-lapse videos, we demonstrate that there are no significant differences in the speed of actin flow at lamellipodia between any of the investigated conditions (**Fig 4B and 4C**). These results strongly suggest that actin polymerization into branched actin networks is not disturbed by the internal tag in β-actin or the overexpression of the ALFA-Nb-GFP.

To demonstrate that the internal tag does not influence cellular processes that are crucially dependent on proper actin function, we evaluated the ability of ALFA-β-actin cells to proliferate and migrate as compared to parental HT1080 cells. Also, to determine a potential negative impact of the expression of the nanobody, we generated a stable cell line overexpressing the ALFA-Nb-GFP and included this cell line in our analyses. To assess cell proliferation, we performed an MTT assay and observed no differences in the the proliferation rate between ALFA-β-actin and parental HT1080 cells, either in the presence or absence of ALFA-Nb-GFP (**Figs 4D and S12A**). To assess cell migration, we performed a wound closure assay and demonstrate that the migration rate of the ALFA-β-actin cells is comparable to the parental cells, suggesting there are no major defects in migration speed due to the internal ALFA tag (**Fig 4E and 4F**). Moreover, co-expression of the ALFA-Nb-GFP has no effect on the migration speed (**S12B and S12C Fig**), indicating that both the tag as well as the nanobody do not influence cell migration under these conditions.

Together, these results indicate that actin dynamics as well as major actin-dependent cellular functions are largely unaffected when actin is tagged at postion T229/A230. Moreover, our findings strongly suggest that ALFA-β-actin can be targeted by fluorescent fusions of the ALFA nanobody without affecting these cellular readouts.

## Tagged actin variants recapitulate differential isoform distribution in mammalian cells and yeast

So far, our results strongly suggest that the T229/A230 position in actin is permissive for epitope tag integration. We term this internal tagging strategy "IntAct" and propose that it can be used to study the molecular principles of actin isoform specificity in biological processes across species. First, to demonstrate the usability for all mammalian actin isoforms, we overexpressed ALFA-tagged versions of the 5 other mammalian isoactins, besides β-actin, in HT1080 and RPE1 cells. This showed that all isoforms colocalized with the actin cytoskeleton (**S13 Fig**), indicating that all tagged isoactins are co-polymerized into actin filaments of various actin-based structures.

Next, to demonstrate that the tagged actins recapitulate the behavior of wild-type isoforms, we sought to investigate the distribution of cytosolic isoforms in HT1080 cells. For this, we used both the ALFA-β-actin cells and an ALFA-γ-actin knock-in cell line that we generated using CRISPR/Cas9. Since it has been shown before that β- and γ-actin display a differential

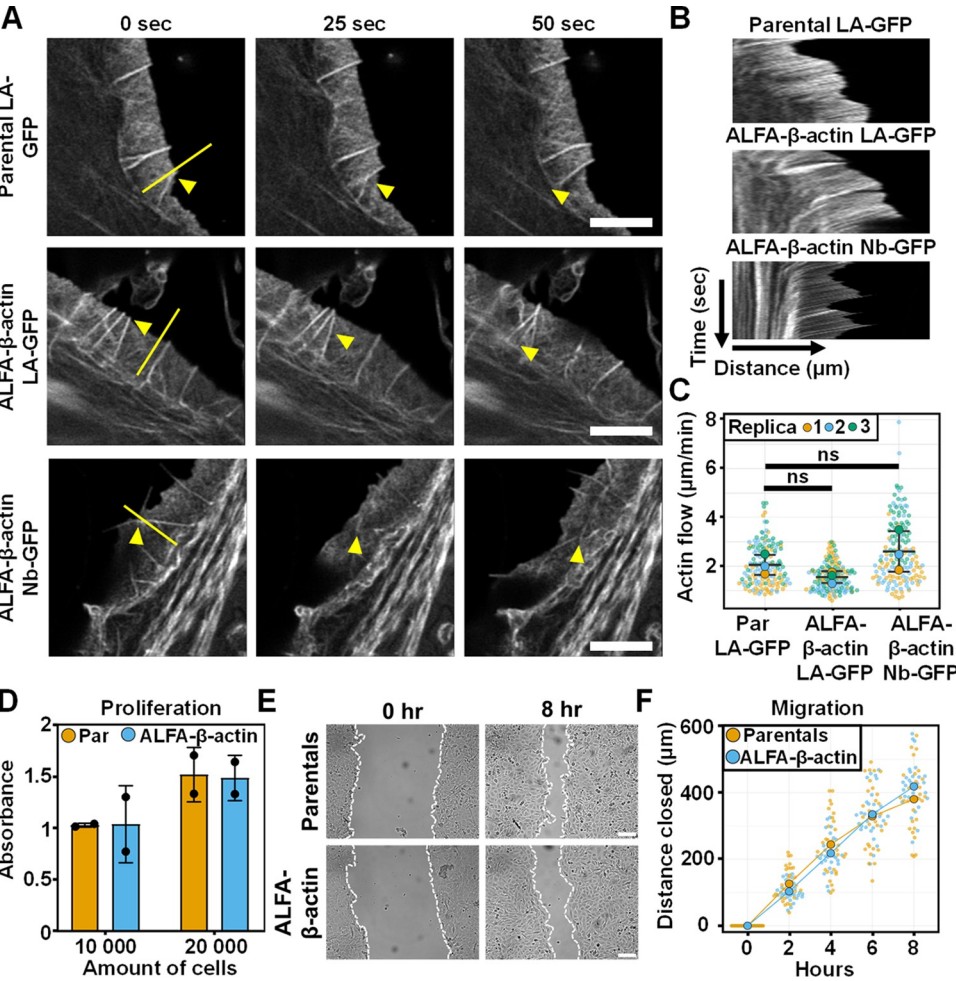

**Fig 4. Actin retrograde flow and cell proliferation and migration are not affected by β-actin internal tagging.** (**A**) Representative airyscan images of HT1080 parental cells transfected with Lifeact-GFP (LA-GFP), HT1080 ALFA-β-actin cells transfected with Lifeact-GFP, and HT1080 ALFA-β-actin cells transfected with ALFA-Nb-GFP. Shown are 3 stills at time point 0, 25, and 50 s and the yellow triangles indicate actin features that display retrograde flow. Yellow line indicates the position of kymographs shown in **B**. The full movies are available as **S2–S4 Movies**. Scale bar: 4 μm. (**B**) Representative kymographs of parental-LA-GFP, ALFA-β-actin-LA-GFP, and ALFA-β-actin-Nb-GFP as indicated by the yellow line in **A**. (**C**) Quantification of the actin retrograde flow (μm/min) in parentals-LA-GFP, ALFA-β-actin-LA-GFP, ALFA-β-actin-Nb-GFP. Large data points represent the average for 3 independent experiments and the small data points represent individual kymographs. The bars show the median and the error bars represents standard deviation. Statistical analysis was performed using unpaired Welch's *t* test. (**D**) Quantification of an MTT proliferation assay performed on parental and ALFA-β-actin HT1080 cells. Individual data points represent the average for 2 independent experiments. Bars represent the mean value and error bars represent standard deviation. (**E**) Representative widefield images of parental and ALFA-β-actin HT1080 cells at time point 0 h and 8 h after scratch induction. Scale bar: 30 μm. (**F**) Quantification of the wound healing assay shown in **E** indicating the distance closed in μm over time in parental (Par) and ALFA-β-actin HT1080 cells. Large data points represent the mean of 3 independent experiments and the individual data points represent the quantification of the individual images. Ten images per condition were acquired per experiment. The numerical data underlying this figure can be found in **S1 Data**.

cellular distribution [10,39], we evaluated whether the localization of these isoforms is similar in parental and IntAct HT1080 cells. For this, we seeded the cells on 2 different substrates. First, we used standard coverslips and second, we used 2D crossbow-shaped micropatterns to normalize the actin organization across cells. Following cell adhesion and fixation, we labeled parental cells with isoform-specific antibodies, and the IntAct cells with the ALFA nanobody

as well as phalloidin for normalization. Cells were imaged with Airyscan super-resolution microscopy and the distribution of actin isoforms in stress fibers, lamellipodia, and filopodia was evaluated through ratio calculations (**Figs 5A–5C and S14**). In line with published observations [10,39], we found on both the glass coverslips as well as the micropatterns, that in parental HT1080 cells, filopodia and lamellipodia display a relatively higher β-/γ-actin ratio as compared to stress fibers (**Figs 5A–5C and S14**). More importantly, we noted a very similar differential distribution of the isoactins in the IntAct HT1080 cells, with β-actin being more enriched in lamellipoda and filopoda as compared to stress fibers (**Figs 5A–5C and S14**). Again, this observation was consistent for cells that were seeded on coverslips or on 2D micropatterns. Notably, we also frequently observed that the β-/γ-actin ratio was even higher in filopodia as compared to lamellipodia but this observation was not consistent enough to reach significance in all conditions. Collectively, the similar β-/γ-actin ratios that we observed for both tagged as well as WT actins in actin-based structures strongly suggest that the differential localization of the cytosolic isoactins β- and γ-actin remains preserved after internal tagging at position T229/A230.

Next, to address if our labeling strategy can be applied to label actin variants in other species, we constitutively expressed IntAct ALFA-actins from yeast (*Saccharomyces cerevisiae* (ALFA-Sc-actin) and *Schizosaccharomyces pombe* (ALFA-Sp-actin)) and human ALFA-β-actin and ALFA-γ-actin (**S15 Fig**) as an additional copy in *S. cerevisiae* and *S. pombe* strains that constitutively express an mNeonGreen-ALFA nanobody fusion protein (ALFA-Nb-mNG) [40] (**S16A and S16B Fig**). We also tried to create budding yeast strains with IntAct as the sole actin copy but unexpectedly, Sc-IntAct could not restore viability in *Δact1* strain in the absence of native actin which was shuffled out using 5′-FOA media (**S17A Fig**). Spot assays of yeast co-expressing IntAct actins and ALFA nanobody showed similar growth rates as wild type (**S17B Fig**). Live-cell imaging in budding yeast revealed that all IntAct actins displayed dynamic foci-like localization abundant in the growing bud and moved to the mother-bud neck before cytokinesis (**S18 Fig**). This localization pattern resembles that of cortical actin patches in *S. cerevisiae* [41,42], which was confirmed by the fact that the foci were positive for F-actin and disappeared after CK666 treatment (**S19A–S19B Fig**). Similarly, live-cell imaging of *S. pombe* cells also revealed a dynamic foci-like localization concentrated at the poles of the cell and at the cell equator (division site) during cytokinesis (**S20 Fig**), identical to known behavior of actin patches in *S. pombe*. Collectively, these observations demonstrate that IntAct proteins incorporate into the Arp2/3-nucleated branched actin patches in both *S. cerevisiae* and *S. pombe*.

To understand how expression of IntAct affects actin dynamics at patches in live cells, we used Abp1-3xmCherry as a patch marker and used its residence time as a quantitative readout for patch lifetime (**S21A Fig**). We then performed time-lapse microscopy and found that actin patch lifetime was the same between *wildtype* cells and cells exogenously expressing either native Sc-Act1 or ALFA-Sc-actin in the presence or absence of ALFA-Nb-mNG (**S21B Fig**). We therefore conclude that IntAct expression did not alter yeast actin patch dynamics. These findings highlight the nondisruptive nature and potential of IntAct as a direct visualization tool for real-time actin dynamics at yeast actin patches alongside exisiting surrogate markers like Abp1 and Arc15 [43,44].

Notably, we did not observe mNG signal into unbranched cytoplasmic actin cables and actomyosin rings (**S18 and S20 Figs**), which are nucleated and elongated by formins (Bnr1 and Bni1 in budding yeast; For3 and Cdc12 in fission yeast). To determine if this was due to an excess of ALFA-Nb-mNG, we generated a budding yeast strain with a galactose inducible-control of ALFA-Nb-mNG expression co-expressing ALFA-Sc-actin. We imaged cells after 2 h of galactose induction, but despite the low levels of ALFA-Nb-mNG at this early time point, we could not detect IntAct signal in actin cables, and only detected signal at actin patches

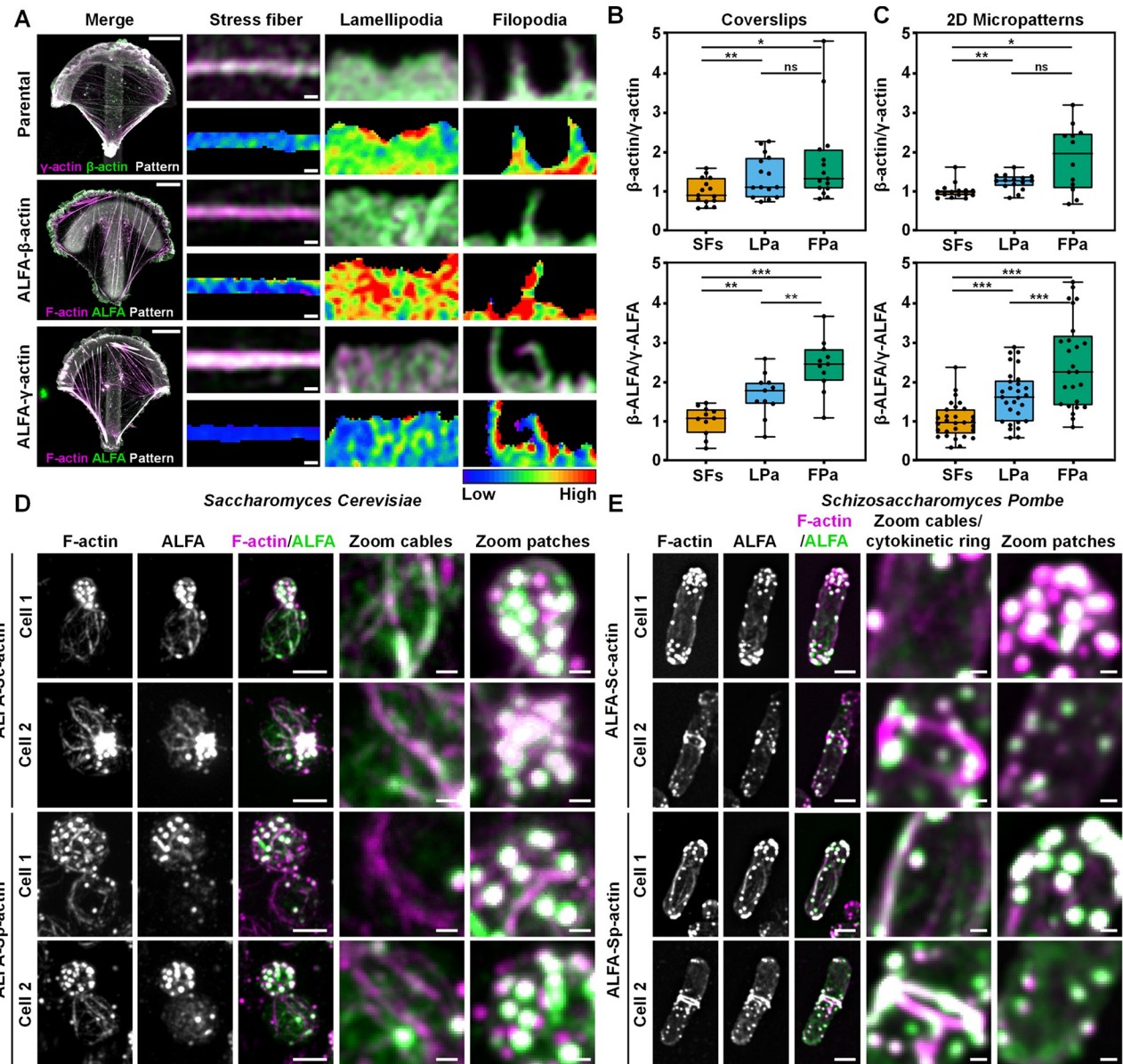

**Fig 5. Tagged actin variants recapitulate differential distribution in mammalian cells and yeast.** (**A**) Representative Airyscan images of cells seeded on 2D crossbow-shaped micropatterns. Shown are parental HT1080 cells stained for β-actin (green) and γ-actin (magenta) and ALFA-β-actin and ALFA-γ-actin cells stained for ALFA tag (green) and F-actin (magenta). The zoom images present stress fibers, lamellipodia, and filopodia, and the 32-color ratio images indicate the ratio between β- and γ-actin (parental cells) or the ratio between β- or γ-actin and total actin. Scale bar: 10 μm. Scale bar zoom images: 0.5 μm. (**B, C**) Quantification of the β-/γ-actin ratio on (**B**) standard coverslips (**S14 Fig**) and (**C**) crossbow-shaped micropatterns of HT1080 parentals and ALFA-β-actin and ALFA-γ-actin cells. Individual data points represent the average ratio per cell calculated by measuring at least 10 regions of interest for stress fibers (SFs), lamellipodia (LPa), and filopodia (FPa). In total, at least 15 cells and <150 adhesions per adhesion category were analyzed over 2 independent experiments. Box plots indicate median (middle line), 25th, 75th percentile (box) and minimum and maximum (whiskers). Statistical analysis was performed using a one-way ANOVA post hoc Tukey's multiple comparison test. * $p < 0.05$, ** $p < 0.01$, *** $p < 0.001$. (**D**) Representative montages of budding yeast cells (*S. cerevisiae*) expressing Sc-IntAct and Sp-IntAct. Cells were fixed and stained for F-actin (magenta) and ALFA tag nanobody (green). Scale bar: 3 μm. Scale bar zoom: 0.5 μm. (**E**) Representative montages of fission yeast cells (*S. Pombe*) co-expressing Sc-IntAct or Sp-IntAct and Lifeact-mCherry. Shown are F-actin (magenta) and ALFA tag nanobody (green). Scale bar: 3 μm. Scale bar zoom: 0.5 μm. The numerical data underlying this figure can be found in **S1 Data**.

(**S22 Fig**). These results suggest either that the ALFA-Nb-mNG prevents IntAct incorporation into actin cables or that incorporated IntAct may be unaccessible to ALFA-Nb binding due to steric hindrance by other yeast actin cable-binding proteins.

Since the incorporation of tagged actins into both branched and unbranched actin filaments appeared unperturbed in mammalian cells, we hypothesized that the lack of actin cable and ring visualization could be a result of steric hindrance between the ALFA-Nb-mNG and the yeast formin proteins rather than a result of the internal tag itself. To test this, we expressed IntAct actins in a wild type *S. cerevisiae* and *S. pombe* strain and assessed their incorporation into the actin cytoskeleton in the absence of the ALFA-Nb-mNG. In *S. cerevisiae*, although the level of incorporation into actin cables varied among the actin variants, we observed incorporation of IntAct actins into both cortical actin patches and cytoplasmic actin cables in the bud (nucleated by formin Bni1) and mother (nucleated by formins Bnr1 and Bni1) cytoplasm (**Figs 5D and** S23A). Specifically, ALFA-Sc-IntAct displayed the greatest degree of cable signal, while ALFA-Sp-IntAct showed no detectable localization to actin cables. Furthermore, ALFA-γ-actin showed weak but discernible localization to actin cables while ALFA-β-actin only showed localization to cortical actin patches. Interestingly, we observed a similar trend in *S. pombe* where ALFA-β-actin, ALFA-γ-actin, and ALFA-Sc-actin incorporated only into actin patches while the ALFA-Sp-actin consistently incorporated into patches, cables, and cytokinetic rings (**Figs 5E and** S23B). Together, these results suggest that it could rather be the association of ALFA-Nb-mNG with the ALFA-actin monomers than the tag itself that is responsible for the lack of incorporation of tagged actins into actin cables when ALFA-Nb-mNG is co-expressed.

Overall, we conclude that internally tagging actin at the T229/A230 position results in a functional actin protein with the ability to incorporate into linear and branched actin filaments in both human and yeast cells, indicating that our IntAct internal tagging strategy can be used to study actin diversity and functionality across species.

## Discussion

In this manuscript, we present a strategy to internally tag actin to study the molecular interactions and dynamics of actin isoforms. At this point, we can only speculate as to why the T229/A230 position seems permissive for manipulation. The T229/A230 residue pair is located in subdomain 4 and is part of a region that has been termed the V-stretch due to the high structural variation that this region exhibits in molecular dynamics simulations of F-actin [45]. To the best of our knowledge, the V-stretch domain has no explicitly described interactions with ABPs, either to G-actin monomers or to F-actin filaments. Specifically, the majority of ABPs that associate with monomeric actin including profilin [46] and gelsolin [47] bind to the barbed end of the actin molecule on subdomains 1 and 3 [48], which is relatively distant from the T229/A230 position. On the other hand, many F-actin associated molecules including myosin [49] and cofilin [50] bind to the so-called outer domain of the actin molecule that comprises part of subdomain 1 and 2 which is also not adjacent to the V-stretch in subdomain 4. Finally, unlike other variable regions such as the D-loop in subdomain 2, the V-stretch is not involved in interactions between monomers in actin filaments [51]. Interestingly, an alanine mutagenesis scan of the entire β-actin protein demonstrated that the V-stretch has a high structural plasticity since the alanine mutants covering this region were not impaired in their folding capacity or their binding to the ABPs DNAse I, adseverin, Thymosin β4, and CAP [52]. Still, it must be noted that the T229/A230 residue pair is very close to the proposed binding interface of the ABP nebulin, and therefore, possibly also the related protein nebulette [53]. Yet, since nebulin and nebulette are exclusively present in skeletal and cardiac muscle, respectively, we expect this will not be an obstacle for studying the nonmuscle and smooth muscle actin isoforms. Future investigations using internally tagged α-skeletal or α-cardiac actin, however, need to carefully control for the possibility that the binding with nebulin or nebulette is influenced by the use of the T229/A230 residue pair.

A previous attempt to internally tag actin with a tetracysteine tag in yeast demonstrated that, out of 8 different internal sites, only the S232/S233 position allowed weak assembly of actin into filament cables and no assembly into rings [31]. We demonstrate here that IntAct actins are integrated into budding and fission yeast actin cables and cytokinetic rings. The extent of integration depends on the species and isoform variant, with native Intact variants consistently displaying the greatest incorporation, as expected. The differential incorporation into actin structures shown by different IntAct actins from yeast and human origin is likely an outcome of hindered interaction with key ABPs arising due to evolutionary divergence from the actin protein sequence of *S. cerevisiae* and *S. pombe*. Similar isoform and species-dependent differences have been demonstrated in a recent study where various actin proteins showed differential recruitment to linear and branched actin structures upon expression in *S. cerevisiae* [54]. For follow-up studies, it would be interesting to use IntAct actins to dissect the differential binding partners and molecular mechanisms that explain these differences.

The fact that the internal position identified by us is extremely close to the S232/S233 residue pair that was previously identified indicates that this region in the actin molecule is relatively permissive for manipulation. Interestingly, in our screen, we also included the S232/S233 residue pair as well as the other 2 adjacent positions, i.e., A230/A231 and A231/S232 and studied their localization to the F-actin cytoskeleton. Although these internally tagged variants were relatively well-integrated into the cytoskeleton with a high colocalization with lamellipodia and filopodia, they were not as well assembled into stress fibers as the T229/A230 variant. This aligns well with the previously published results in yeast [31] and suggests very specific structural requirements for actin internal tagging. Importantly, the differential integration of the internally tagged actins into lamellipodia, filopodia, and stress fibers may also explain some of the large differences in colocalization that we observed between the 2 investigated cell types (HT1080 and RPE1). RPE1 cells have relatively more stress fibers as compared to HT1080 cells and since the A230/A231 and A231/S232 internally tagged actins are mostly integrated into lamellipodia and filopodia, this may explain their low colocalization in RPE1 cells.

We show that in cells from human and yeast, IntAct actins are integrated in F-actin-based structures that are composed of branched filaments such as lamellipodia and actin patches as well as unbranched filaments such as filopodia, stress fibers, and actin cables. For lamellipodia, we also demonstrate that rearward treadmilling, which is indicative for polymerization speed, is not different between cells that express wild type or IntAct β-actin. It is well established that the assembly of branched actin filaments is controlled by the Arp2/3 complex, while the assembly of unbranched filaments is controlled by formins [55]. We therefore conclude that internal tagging of actin at position T229/A230 does not prevent the association with these different classes of actin nucleating and elongation factors, something which is supported by the observed binding of formins to IntAct β-actin. Although our findings strongly suggest that the nucleation and polymerization of actin is largely unaffected, future in vitro reconstitution experiments will be required to precisely dissect to what extent the biochemical properties of IntAct actins are similar to those of wild-type actins. For this, recent developments in actin production and purification systems such as Pick-ya actin could be very instrumental [56,57].

An important difference between our results in yeast and human cells is the ability to visualize the ALFA-tagged actins in living cells. In human cells, the nanobody allowed visualization of various types of actin-based structures including filopodia, lamellipodia, and stress fibers while in yeast, only the branched actin patches were visible, and the visualization of unbranched actin cables or actomyosin rings was hampered when the ALFA-Nb-mNG was constitutively expressed or induced to low expression levels. Our inability to visualize actin cables and rings currently represents a limitation for live-cell imaging of these structures in yeast and it remains to be determined whether this discrepancy is due to improper

incorporation of ALFA-actin in the presence of the nanobody or because of steric hindrance of the ALFA-Nb-mNG to localize to actin cables or rings. Our finding that ALFA-tagged actin can be integrated in unbranched actin cables and rings in the absence of the nanobody at least indicates that the tag itself is likely not the reason for our inability to visualize unbranched actin in living yeast cells. To overcome this limitation and allow future live-cell imaging experiments to study yeast actin cables and rings, optimization of other small tag/fluorophore combinations is required. One interesting possibility could be the use of ALFA tag nanobodies with a higher dissociation rate in order to allow actin monomer integration in the unbound state [58]. Furthermore, it should be noted that in this study, we have established only 1 small tag/fluorophore combination for live cell targeting of the tag in living cells, namely the ALFA tag. While our findings extend the applicability of the ALFA-tag system to visualize homologous proteins with known tagging issues, for future experiments that require simultaneous targeting of 2, or more, isoactins in the same cell, additional tag/fluorophore combinations should be optimized. For this, novel bottom-up design approaches for nanobody-based tags, similar to the ALFA tag design [35], are required.

In conclusion, our results demonstrate that the T229/A230 residue pair allows internal tagging of actin, which can be used to study the localization, dynamics, and molecular interactors of specific actin isoforms. Our novel internal tagging system complements the already existing repertoire of actin probes such as N-terminal fusions, Lifeact, and F-Tractin and allows researchers to study so far unknown properties of specific actin variants without the necessity to manipulate the N-terminus. Since our approach also allows intracellular targeting of specific actin variants in living cells, we envision that fusing the ALFA nanobody with peroxidases (e.g., APEX2 [59]) or biotin ligases (e.g., TurboID [60]) will allow the future investigation of isoactin-specific molecular interactions of both monomeric and filamentous actin. Importantly, we also demonstrate the possibility of tagging actin at position T229/A230 in yeast actins, suggesting that our IntAct approach can be applied across species to study differences in isoactins. Finally, we envision that, by tagging mutant actin variants, our approach could also open avenues to unravel the disease-causing mechanisms of a wide variety of actinopathies [4], for which currently no strategy is available.

## Materials and methods

### Cell culture

HT1080 fibrosarcoma cells were cultured in 1× DMEM + 4.5 g/L D-Glucose, NEAA (Gibco, Lot#2246377) and supplemented with 10% (vol/vol) fetal bovine serum (FBS), 1× Glutamax (Gibco, 2063631), 1 mM Sodium Pyruvate (Gibco, 2010382), and 0.5× Antibiotic-Antimycotic (Gibco, 15240–062). RPE1 cells were cultered in advanced DMEM/F-12 + nonessential amino acids + 110 mg/L Sodium Puruvate (Gibco, Lot#12634010) supplemented with 10% (vol/vol) FBS and 1× Glutamax (Gibco, 2063631). All cell lines were cultured and kept at 37˚C with 5% $CO_2$.

### Yeast strains

All yeast strains used in this study are listed in the Supporting information (**S1 Table**). Yeast strains were constructed using protocols previously described [61].

### Antibodies and reagents

The following primary antibodies were used: anti-β-actin (#MCA5775GA, Bio-Rad Laboratories, 1:100 for immunofluorescence, 1:1,000 for western blot), anti-β-actin arginylated (N-

terminal) (#ABT264, EMD Millipore, 1:1,000 for western blot), anti-γ-actin (#MCA5776GA, Bio-Rad Laboratories, 1:100 for immunofluorescence, 1:1,000 for western blot), anti-Flag (#F1804-1MG, Sigma Aldrich, 1:1,000 for western blot), anti-AU1 (#NB600-452, Novus Biologicals, 1:100 for immunofluorescence), anti-AU5 (#NB600-461, Novus Biologicals, 1:100 for immunofluorescence), anti-profilin (#3246, Cell Signaling, 1:1,000 for western blot), anti-cofilin (#5175P, Cell Signaling, 1:1,000 for western blot), anti-DIAPH1 (#610848, BD Science, 1:1,000 for western blot), anti-FMNL2 (#HPA005464, Sigma Aldrich, 1:1,000 for western blot), anti-Myosin IIA Heavy chain (#909801, Biolegend, 1:100 for immunofluorescence, 1:1,000 for western blot), anti-α-actinin (#A5044, Sigma Aldrich, 1:1,000 for western blot), anti-Lamin AC (#MA1-06102, Invitrogen, 1:1,000 for western blot), anti-tubulin (clone E7, in-house hybridoma 1:4,000 for western blot), anti-aSMA (#19245S, Cell Signaling, 1:1,000 for western blot). Actin was stained with anti-actin (#A2066, Sigma Aldrich, 1:4,000 for western blot), Alexa-488-conjugated phalloidin or Alexa-568-conjugated phalloidin (Life Technologies, 1:200), ALFA tag was stained with HRP-conjugated sdAb anti-ALFA primary antibody (#N1505, NanoTag Biotechnologies, 1:1,000 for western blot), anti-ALFA-atto488 conjugate or anti-ALFA-Alexa647 conjugate (#N1502 NanoTag Biotechnologies, 1:100 for immunofluorescence). Secondary antibodies conjugated to Alexa647, Alexa568, or Alexa 488 were used (Life Technologies, 1:400 for immunofluorescence).

## Generation of overexpression constructs

All overexpression constructs with the internally tagged actins were generated by Gibson assembly (New England Biolabs). Briefly, 2 PCRs were performed per construct to generate a DNA fragment upstream of the tag and a fragment downstream of the tag. Primers used for the PCR reactions are given in the Supporting information (S2 Table). Both fragments contained the DNA coding for the tag which functions as an overlapping sequence in the Gibson assembly reaction. The pcDNA3.1 vector backbone was linearized using restriction enzymes HindIII and NheI and 100 ng of vector was used in every Gibson assembly. PCR fragments were added in a 1:6 vector:insert molar ratio and all assembly reactions were incubated for 50 degrees for 50 min. Half the product was transformed into Top10 competent bacteria and clones were screened for the correct vectors.

## Generation of knock-in cell lines

gRNAs and HDR templates used for the generation of FLAG-, AU1-, AU5-t-, and ALFA-knock-in cells are given in the Supporting information Tables (S3 Table). Lipofectamine 2000 (Thermo Fisher, ref. 11668027) was used to transfect the HT1080 cells. To increase the efficiency of the knock-in approach, we applied a co-selection procedure using ouabain as described previously [36]. The gRNA and HDR template for mutating the ATP1A1, which leads to ouabain resistance, are given in the Supporting information Tables (S3 Table). Flow cytometry for the respective tags was performed 1 week after the initial transfection to determine the number of positive cells. Subsequently, treatment with 0.75 μM ouabain was started and after 2 weeks, single clones were generated from the selected cells. Positive clones were selected based on intracellular FACS staining and were further used for immunofluorescence and western blot.

## Generation of yeast constructs

All plasmids used in this study are listed in the Supporting information (S4 Table). The IntAct DNA sequence for *Saccharomyces cerevisiae* (Sc-IntAct) and *Schizosaccharomyces pombe* (Sp-IntAct) were synthesized by GeneArt (Life Technologies). Sequences for human β-actin IntAct

and human γ-actin IntAct were amplified from plasmids used for overexpression. All plasmids were constructed using NEBuilder Hifi (E2621, New England BioLabs).

## Immunofluorescence mammalian cells

All steps were performed at room temperature. Cells were seeded on coverslips and fixed using 4% PFA for 10 min. Permeabilization was performed with 0.1% Triton X-100 for 5 min. After washing with 1× PBS, samples were blocked with 20 mM PBS/glycin. Primary antibody incubation was performed for 1 h. Subsequently, samples were washed 3 times with 1× PBS and incubated with the appropriate secondary antibody for 1 h in the dark. The samples were then washed twice with 1× PBS and once with MilliQ. Lastly, the samples were sealed in Mowiol and dried overnight.

## Immunofluorescence and phalloidin staining of yeast

Immunofluorescence was performed using a modified version of a protocol described previously [62]. Briefly, yeast strains were grown overnight at 25˚C in YPD broth/EMM-Ura. The overnight culture was diluted and allowed to grow until mid-log phase. Cells were fixed with 4% formaldehyde for 60 min at 25˚C, washed twice with 1× PBS, and finally resuspended in 200 μl of 1.2M Sorbitol Phosphate-Citrate (SPC) buffer (1.2 M Sorbitol, 1 M K2HPO4, 1 M Citric acid); 25 μl of Long-Life Zymolase (#786–036, GBiosciences) was added to digest the budding yeast cell wall (Lysing enzymes from *Trichoderma harzianum* (#L1412, Sigma-Aldrich) was also added to a final concentration of 1 mg/ml for fission yeast) and the suspension was incubated with mild shaking at 37˚C for 60 min. The cells were then washed twice with ice-cold SPC buffer and incubated with 500 μl of blocking buffer 2% bovine serum albumin (BSA) + 0.1% Triton X-100 in PBS at room temperature for 15 min with shaking. The cells were pelleted and resuspended in 500 μl of Antibody Dilution Buffer (1% BSA + 0.05% Triton X-100 in PBS) containing FluoTag-X2 anti-ALFA-Alexa647 (#N1502, Nanotag Biotechnologies) at a final dilution of 1:500. The cell suspension was then incubated overnight with rotation at 4˚C. Next day, cells were washed twice with 1× PBS and finally resuspended in 20 μl of 1× PBS, and 5 μl of the final cell suspension was mounted on a glass-bottom dish coated by poly-L-lysine (#P4707, Sigma Aldrich). Phalloidin staining of yeast actin structures was done as per previously described protocols [63,64]. Briefly, cells were grown at 25˚C till mid-log phase and fixed with 4% paraformaldehyde. The cells were washed thrice with 1× PBS and labeled phalloidin was added to a final concentration of 0.4 μM (in 50 μl of 1× PBS) and the tubes were kept in a rotating shaker overnight at 4˚C. The cells were washed twice with 1× PBS on the next day and seeded on a concanavalin A coated glass-bottom dish. Images were acquired using Andor Dragonfly 502 spinning-disk confocal setup (Oxford Instruments, United Kingdom) consisting of a Leica Dmi8 fully motorized inverted microscope equipped with Andor Sona scMOS camera or Olympus FV3000 point-scanning confocal setup consisting of an Olympus IX83 fully motorized inverted microscope equipped with high-sensitivity GaAsP photomultiplier tube (PMT) detectors.

## Micropatterning

Crossbow-shaped micropatterns were patterned onto 3.5 cm glass-bottom dishes with a 20 mm glass observation window (D-35-20-1.5N, Cellvis) using maskless photopatterning as described before [65]. Micropatterning was performed using a PRIMO photopatterning system with a 200 mW 375 nm laser (Alvéole), installed on a Nikon Ti2 inverted microscope stand with stage-up kit, motorized stage, hardware autofocus system (PFS, Nikon), and equipped with a Hamamatsu Orca Flash4.0 LT sCMOS camera, and CFI S Plan Fluor ELWD

20× objective (Nikon). NIS-Elements (version 5.30.02, Nikon) with Leonardo plugin (version 4.18.6594, Alvéole) was used for controlling the microscope and PRIMO patterning unit. Before micropatterning, glass-bottom dishes were first treated with a handheld Corona treater (ETP, Model BD-20AC, laboratory Corona Treater) for approximately 3 min. Immediately after corona treatment, 200 μl poly-d-lysine (PDL) (P7405, Merck) at 0.5 mg/ml in 50 mM K-HEPES (pH 8.3) was added to the center of each dish, followed by incubation for 30 min at room temperature. PDL solution was then carefully aspirated, and dishes were washed 3 times with 1 ml 100 mM K-HEPES (pH 8.3). Immediately after washing, 200 μl of mPEG-SVA solution (50 mg/ml mPEG-SVA, MW 5000 (Laysan Bio) in 100 mM K-HEPES (pH 8.3)), prepared <1 min before use, was added to the center of each dish, followed by incubation at room temperature for 30 min. Dishes were then washed 5 times with sterile ddH$_2$O and dried completely. For micropatterning, 3 μl of PLPP photo-initiator gel (Alvéole) in 50 μl 100% ethanol was added to the center of each dish and spread evenly over the mPEG-coated glass surface by gently rotating it. Dishes were then placed inside a dark box (with lid ajar), and PLLP gel was left to dry for approximately 15 min, after which the PLPP-coated dishes were immediately used for photopatterning (or within 3 h). Photopatterning was performed on the PRIMO system (calibrated just before use) using a 20× objective and UV dose set to 90 mJ/mm$^2$. Approximately 15.000 arrow shapes were patterned per dish. After photopatterning, dishes were washed extensively with ddH$_2$O to remove the PLPP film, followed by incubation with either gelatin alone (2% (w/v) solution), or in combination with fluorescent fibrinogen-A647 (F35200, Thermo Fisher Scientific) at 10 μg/ml for 30 min and washed 3 times with PBS.

## Imaging

Imaging was performed on a Zeiss LSM900 laser scanning confocal microscope and a Leica DMI6000 epifluorescence microscope. Images on the LSM900 were acquired using a 63× 1.4 NA oil objective. Alexa488 was excited at 488 nm and emission light was detected between 490 and 575 nm. Alexa568 was excited at 561 nm and emission was detected between 555 and 700 nm. Alexa647 was excited at 640 nm and emmission was detected between 622 and 700 nm. Raw images were processed using the Zeiss Zen 3.1 blue edition software. Images on the Leica DMI6000 were acquired with an HC PL APO 63× 1.40 NA oil objective and a metal halide lamp. Alexa488 was excited between 450 and 490 nm and emmision light was detected between 500 and 550 nm. Alexa568 was excited between 540 nm and emmision light was detected between 567 and 643 nm.

## Live-cell imaging mammalian cells

Parental and/or ALFA β-actin HT1080 cells were seeded in WillCo wells (WillCo Wells B.V.). The next day, cells were transfected with Lifeact-GFP, ALFA-Nb-GFP, or ALFA-Nb-mScarlet together with Lifeact using Lipofectamine 2000 (Invitrogen, lot#1854327). Prior to imaging, DMEM was replaced by imaging medium (HBSS, Ca/Mg, 5% FCS, 25 mM HEPES) and incubated for approximately 10 min. Live-cell imaging was performed using a Zeiss LSM880 with Airyscan and data was acquired using a 63× 1.4 NA oil objective. During single color live-cell imaging, Lifeact-GFP and ALFA-Nb-GFP was excited using a mass beam splitter (MBS) 488 and emission light was collected using a 495 to 550 band pass/570 long pass (LP) filter. Sequentially imaging was done for Lifeact-GFP together with ALFA-Nb-mScarlet. Lifeact-GFP and ALFA-Nb-mScarlet were excited using an MBS 488/561 and emission light was collected using a BP 420–480/BP 495–550 for Lifeact-GFP and emission light was collected using a BP 570–620 + LP 645 for ALFA-Nb-mScarlet. Time series were collected with a frame interval of 5 s for actin flow and 15 s for colocalization of Lifeact with ALFA-Nb-mScarlet. Raw images were

processed using the Zeiss Zen 2.1 Sp1 software. Image series were analyzed using ImageJ and the PCC was calculated using the ImageJ tool Coloc2 (PSF: 3, Costes randomizations 10).

## Live-cell imaging of yeast cells

Yeast cells were grown at 25°C to mid-log phase in Synthetic Complete (SC) medium and plated on a glass-bottom dish (#D35C4-20-1.5-N, Cellvis) coated by 6% Concanavalin A (C2010, Sigma-Aldrich). Fission yeast cells were grown at 30°C to mid-log phase in EMM-Ura broth and adhered on an agarose pad for live-cell imaging. For the induction of ALFA-Nb-mNG, cells were grown in SC+3% Raffinose and galactose was added to a concentration of 2% before start of imaging. For CK666 (SML0006, Sigma-Aldrich) treatment, DMSO (solvent control) or CK666 was added to a final concentration of 200 μM in the media contained in the glass-bottom dish and the prepared dish was taken for live-cell imaging. For actin patch life-time measurement, single-plane images were acquired at 500 ms interval for 2 min and actin patch lifetime was calculated as the residence time of Abp1-3xmCherry signal. Images were either acquired using Andor Dragonfly 502 spinning-disk confocal setup (Oxford Instruments, UK) consisting of a Leica Dmi8 fully motorized inverted microscope equipped with Andor Sona scMOS camera or Olympus FV3000 point-scanning confocal setup consisting of an Olympus IX83 fully motorized inverted microscope equipped with high-sensitivity GaAsP PMT detectors.

## Western blot

For western blot, cells were lysed with 2× Laemmli (5 ml 0.5 M Tris (pH 6.8), 8 ml 10% SDS, 4 ml Glycerol, few grains bromophenol blue, 2 ml 2-mercaptoethanol, and 1 ml MilliQ). Samples were loaded onto 10% or 15% SDS-PAGE gels for separation. Separation was accomplished by running for approximately 2 h at 100 V in 1× Running buffer (100 ml 10× TBS, 10 ml 10% SDS, and 900 ml MilliQ). Proteins were transferred to PVDF membranes for approximately 1 h at 100 V in transfer buffer (100 ml 10× TBS, 200 ml MeOH, and 700 ml MilliQ). Membranes were blocked in 5% milk (ELK milk powder, Friesland Campina) in TBST (1× TBS, 0.1% Tween20) and incubated with primary antibodies overnight at 4°C while rotating. After 3 times washing with 1× TBST, membranes were incubated in the dark for 1 h with secondary antibodies while rotating. Washing with 1× TBST was repeated and subsequently, the protein bands were visualized using a Typhoon FLA 7000 (GE Healthcare). ImageJ was used to analyze the protein bands. For yeast cell lysates, TCA precipitation method was used as described [40]; protein was probed with HRP-conjugated sdAb anti-ALFA primary antibody (#N1505, Nano-Tag biotechnologies) or anti-actin monoclonal antibody (#MA1-744, Invitrogen).

## F-/G-actin ratio

Cells were seeded and the next day washed in ice-cold PBS and lysed on ice for 10 min with F-actin stabilization buffer (0.1 M PIPES (pH 6.9), 30% glycerol, 5% DMSO, 1 mM MgSO4, 1 mM EGTA, 1% Triton X-100, 1 mM ATP, protease inhibitor cocktail (Sigma-Aldrich, 11697498001)). The cells were harvested and spun down for 10 min at 1,000 g and 4°C. The supernatant was collected and spun down at 16,000 g for 75 min at 4°C to separate the G- and F-actin fractions. The supernatant, containing G-actin, was collected and the pellet, containing F-actin, was solubilized in depolymerization buffer (0.1 M PIPES (pH 6.9), 1 mM MgSO4, 10 mM CaCl2, 5 μM cytochalasin D, 1% SDS) or 2× Laemmli (5 ml 0.5 M Tris (pH 6.8), 8 ml 10% SDS, 4 ml Glycerol, few grains bromophenol blue, 2 ml 2-mercaptoethanol, and 1 ml MilliQ). For the negative control, cells were treated with 1 μM Latrunculin A 30 min before lysis to disrupt F-actin. The F-/G-ratio was determined by western blot analysis.

## ALFA tag co-immunoprecipitation

Cells were seeded and the following day washed with ice-cold PBS. Cells were lysed with ice-cold lysis buffer (10 mM Tris, 150 mM NaCl, 2 mM MgCl2, 2 mM CaCl2, 1% Brij-97), supplemented with 1× protease inhibitor cocktail (Sigma-Aldrich, 11697498001). Cell lysates were centrifugation at 16,000 g for 1 h at 4˚C. For ALFA tag pulldowns, ALFA-Selector ST beads (NanoTag Biotechnologies, N1510) were washed with lysis buffer. For input, 4% or 5% of the clarified lysate was collected as positive control and diluted in 2× Laemmli buffer. The rest of the sample was combined with the beads and incubated for 1 h at 4˚C with rotation. Next, the beads were pelleted by centrifugation for 1 min at 1,000 g. The beads were washed with lysis buffer supplemented with 0.1 mM PMSF (Sigma-Aldrich, P7626-5G) and eluted for 20 min with 2× Laemmli buffer containing 0.2 µM elution peptide (NanoTag Biotechnologies, N1520-L) at RT while shaking. The samples were centrifuged for 1 min at 3,000 g and the supernatant collected as elute sample.

## FLAG co-immunoprecipitation

The day before the co-immunoprecipitation, BSA- and IgG-coated dynabeads for pre-clearing were prepared by mixing dynabeads slurry (40 µl per sample) with 500 µl PBS/3% BSA or 500 µl PBS with 2.5 µg mouse IgG1 (BioLegend, 400102), respectively. Similarly, BSA-coated dynabeads for the immunoprecipitation were prepared by mixing dynabeads slurry (60 µl per sample) with 500 µl PBS/3% BSA. FLAG knock-in clones were seeded and when the cells reached 80% confluency, they were washed once with cold PBS and lysed for 15 min at 4˚C with 1 ml lysis buffer (1% Brij-97, 10 mM Tris-HCl (pH 7.5), 150 mM NaCl, 2 mM MgCl2, 2 mM CaCl2, protease inhibitor cocktail (Sigma-Aldrich, 11697498001)). Cell lysates were collected by scraping and spun down at 16,000 g and 4˚C for 75 min to clarify the lysate. In the meantime, the pre-clear beads were washed once with lysis buffer and resuspended in a total of 40 µl per sample. The supernatant was pre-cleared for 1 h at 4˚C while shaking, with both the BSA- and IgG-coated beads. After pre-clearing, the beads were removed and 40 µl of the supernatant was collected as input. The sample was split into 2 parts, which was supplemented with lysis buffer to a volume of 2 ml, and 6 µg of either mouse anti-FLAG (Sigma, F1804) or mouse IgG1 (BioLegend, 400102) was added, and samples were incubated for 1 h at 4˚C under rotation. Meanwhile, the immunoprecipitation beads were washed with lysis buffer and resuspended in a total of 60 µl per sample. The beads were added, and the samples were incubated for an additional 2 h. After washing 5 times with washing buffer (1% Brij-97, 10 mM Tris-HCl (pH 7.5), 150 mM NaCl, 2 mM MgCl2, 2 mM CaCl2, 1 mM PMSF), the beads were eluted in 100 mM glycine (pH 3.0) for 5 min while rotating. The samples were neutralized by adding 1/10th volume of 1 M Tris-HCl (pH 8.5) and loaded on 15% SDS-PAGE gels for western blot analysis.

## Wound closure assay

Cells were seeded to 100% confluency and a scratch was made with a 200 µl tip from the top to the bottom of the well. After scratching, the cells were washed 1× with PBS and fresh media was added to the cells. The same position was imaged with a Leica DMI6000 epifluorescence microscope after 0 h, 2 h, 4 h, 6 h, and 8 h. The images were analyzed using the Wound healing size tool plugin in ImageJ [66]. First, the edges of the wound were manually annotated and the plugin was subsequently used to measure the average wound width. Using the average wound width, the relative distance closed between every time point and time point 0 was calculated.

## MTT proliferation assay

Cells (10.000 or 20.000) were seeded in a 96-well plate. After 16 h of incubation, media was replaced by media containing the tetrazolium dye MTT (0.45 mg/ml). After 2 h of MTT incubation at 37°C in the $CO_2$ incubator, 150 μl DMSO was added to the cells and incubated for 10 min on an orbital shaker until the crystals were dissolved. The absorbance at 560 nm was determined on a microplate reader (iMark microplate absorbance reader, Bio-Rad).

## Image analysis

All image analysis was performed using ImageJ version 1.53C [67]. PCC was calculated using the ImageJ tool Coloc2 (PSF: 3, Costes randomizations 10). To plot the PCC values, a unity-based normalization was performed, adjusting the raw PCC values to the maximal expected colocalization (the value for N-terminally tagged actin) and the minimal expected colocalization (the value for C-terminally tagged actin) according to Eq 1:

$$\text{Normalized PCC} = \frac{Raw\ PCC - Mean\ PCC_{Cterm}}{Mean\ PCC_{Nterm} - Mean\ PCC_{Cterm}} \qquad [1]$$

For the β-/γ-actin fluorescence intensity ratio analysis in **Figs 5** and **S14**, all images were first corrected by background subtraction. For each cell, at least 10 regions of interest (ROIs, 0.5 μm × 0.5 μm) were selected for stress fibers, lamellipodia, and filopodia and ROI selection was performed based on a single fluorescence channel to avoid bias in the selection of the regions. Moreover, the selected ROIs were entirely contained within an actin-based structure and did not include any signal from other structures. In order to calculate the ratio values for the parental cells, the β-actin channel intensity was divided by the γ-actin channel intensity which resulted in a β-/γ-actin ratio. For the IntAct cells, a double normalization was performed in order to retrieve the β-/γ-actin ratio. For this, first the ALFA-β-actin or ALFA-γ-actin channel intensity was divided by total actin channel intensity (phalloidin), resulting in values for ALFA-β-actin:total actin and ALFA-γ-actin:total actin. To subsequently calculate the β-/γ-actin ratio in actin-based structures, these ratios were divided by each other. Eq 2 was used to calculate the β-/γ-actin ratio in the ALFA IntAct cells:

$$\text{Normalized } \beta - /\gamma - \text{actin ratio} = \frac{\left(I_{\beta-ALFA}\big/I_{\text{Total actin}}\right)}{\left(I_{\gamma-ALFA}\big/I_{\text{Total actin}}\right)} \qquad [2]$$

Myosin IIA spacing was measured on stress fibers by drawing a line along the stress fiber of 7.5 μm, which was saved as an ROI. The ROI was overlaid in the myosin IIA channel and an intensity plot profile was made. Subsequently, the positions of each high intensity peak in the plot profile were saved. The distance between every high intensity peak along a stress fiber was calculated and an average of all the distances was taken as the average myosin IIA spacing along the single stress fiber. This procedure was repeated for at least 30 stress fibers per condition.

## Data visualization

All graphs were designed in GraphPad Prism 9.0 (GraphPad Software) except for **Figs 4C, 4F and S12B** which were generated with SuperPlotsofData [68].

### Statistics

The type of statistical test, *n* values, and *P* values are all listed in the figure legends, in the figures, or in the source data. All statistical analyses were performed using GraphPad Prism or Microsoft Excel, and significance was determined using a 95% confidence interval.

## Supporting information

**S1 Fig. Crystal structure of the actin domains and the position of each residue pair used in the screening.** Crystal structure of uncomplexed globular actin (magenta ribbon, PBD accession number: 1J6Z[32]) indicating subdomain 3, subdomain 4, and the ATP pocket. Zooms show each domain and ATP pocket and their associated position for each distinct residue pair. (TIF)

**S2 Fig. Identification of actin T229/A230 as a permissive target site for epitope tag integration.** Representative widefield immunofluorescence images of F-actin (magenta) and FLAG (green) in HT1080 cells that overexpress the tagged β-actin variants. Shown are the 8 internally tagged variants that are not depicted in **Fig 1B**. Scale bar: 15 μm. Scale bar zoom: 5 μm. (TIF)

**S3 Fig. Identification of actin T229/A230 as a permissive target site for epitope tag integration.** (**A**) Representative widefield immunofluorescence images of F-actin (magenta) and FLAG (green) in RPE1 cells that overexpress the tagged β-actin variants. Shown are 11 internally tagged variants and the N- and C-terminally tagged β-actin. Scale bar: 15 μm. Scale bar zoom: 5 μm. (**B**) Colocalization analysis of the microscopy results in **A** showing the normalized Pearson's correlation coefficient for each of the actin variants. Individual data points indicate single cells and in total, at least 10 cells from 2 independent experiments were included in the analysis. Bars represent the mean value, and error bars represent standard error of mean (SEM). The numerical data underlying this figure can be found in **S1 Data**. (TIF)

**S4 Fig. Flow cytometry, western blot, and Sanger sequencing data of cells having a CRISPR/Cas9-mediated knock-in of FLAG, AU1, AU5, or ALFA tag in β-actin.** (**A**) Flow cytometry data of cells having a CRISPR/Cas9-mediated knock-in of FLAG, AU1, or AU5. Pool of cells was stained for their appropriate tag. (B) Flow cytometry data of cells having a CRISPR/Cas9-mediated knock-in of FLAG, AU1, AU5, or ALFA after selection with Ouabain. Pool of cells was stained for their appropriate tag. (C) Representative western blot of β-actin in parental HT1080 (WT) and 2 independent hemizygous FLAG-β-actin HT1080 clones (FLAG/-). (D) Sanger sequencing result of homozygous ALFA-β-actin, hemizygous FLAG-β-actin, and hemizygous AU5-β-actin HT1080 cells. Highlighted in blue is the ALFA, FLAG, or AU5-t sequence at position T229/A230 in β-actin. Alignment of β-actin sequence, the tag sequence in 1 or 2 alleles and the possible disrupted allele sequence. (TIF)

**S5 Fig. Position T229/A230 is a permissive site for tagging both cytoplasmic isoactins and the expression of isoactins in hemizygous FLAG and homozygous ALFA cells is unaltered.** (**A**) Representative widefield images from cells that have a CRISPR/Cas9-mediated knock-in of FLAG in γ-actin in ALFA-β-actin cells. Cells were labeled for ALFA (magenta) and FLAG (green) staining, respectively, for β-actin and γ-actin. Scale bar: 15 μm. (**B**) Representative western blot showing β-actin and γ-actin expression in parental HT1080 (Par) and hemizygous FLAG-β-actin HT1080 cells. Tubulin was used as a loading control. (C) Quantification of B showing β-actin and γ-actin expression in parental HT1080 and hemizygous FLAG-β-actin

HT1080 cells normalized to tubulin. Individual data points represent 3 independent western blots. Bars represent the mean value and error bars represent standard error of mean (SEM). (D) Representative western blot showing γ-actin expression in parental HT1080 (Par) and homozygous ALFA-β-actin HT1080 cells. Tubulin was used as a loading control. (E) Quantification of the γ-actin expression in ALFA-β-actin and parental HT1080 cells normalized to tubulin. Individual data points represent 2 independent western blots. Bars represent the mean value and error bars represent standard deviation. (F) Representative western blot showing αSMA and ALFA tag expression in parental HT1080 (Par), homozygous ALFA-β-actin HT1080 cells, and myofibroblast (positive control for αSMA). The numerical data underlying this figure can be found in S1 Data.
(TIF)

**S6 Fig. N-terminal arginylation and translocation of tagged β-actin.** (**A**) Representative western blot showing arginylated β-actin and β-actin expression in parental HT1080 (Par), homozygous ALFA-β-actin HT1080 cells, and hemizygous FLAG-β-actin HT1080 cells. Tubulin was used as a loading control. (**B**) Representative western blot of nuclear fractionation assay showing vinculin (cytosol marker), Lamin A/C (nuclear marker), and β-actin expression in parental HT1080 and homozygous ALFA-β-actin HT1080 cells.
(TIF)

**S7 Fig. Stress fibers have a similar architecture in ALFA-β-actin cells as compared to parental cells.** (**A**) Representative Airyscan images of stress fibers from parental and ALFA-β-actin cells stained against F-actin, ALFA tag, and Myosin IIA. Scale bar: 1 μm. (**B**) Quantification of the Myosin IIA spacing on stress fibers in **A**. Bars represent the mean value and error bars represent the standard deviation. In total, the average myosin spacing of at least 30 stress fibers from 15 different cells are included in the analysis. The numerical data underlying this figure can be found in **S1 Data**.
(TIF)

**S8 Fig. ALFA-β-actin integrates properly into filaments in living cells.** (**A**) Representative Airyscan images of HT1080 cells expressing LifeAct-GFP (green) and ALFA-tag Nb-mScarlet (magenta) after 0 min, 10 min, and 25 min. Full movie is available as **S1 Movie**. Scale bar: 10 μm. (**B**) Colocalization analysis of the microscopy results in **A** showing the Pearson's coefficient for each time point. Individual data points indicate single cells and in total, 5 different movies were included in the analysis. Bars represent the mean value and error bars represent standard error of mean (SEM). The numerical data underlying this figure can be found in **S1 Data**.
(TIF)

**S9 Fig. Localization of GFP in IntAct cells and fluorescent ALFA-Nb fusions in parental cells.** (**A**) Representative widefield images of HT1080 parental cells and ALFA-β-actin cells expressing GFP (green) and stained for F-actin (magenta). Scale bar: 10 μm. (**B**) Representative widefield images of HT1080 parental cells and ALFA-β-actin cells expressing ALFA-Nb-GFP (green) and stained for F-actin (magenta). Scale bar: 10 μm. (**C**) Representative widefield images of HT1080 parental cells and ALFA-β-actin cells expressing ALFA-Nb-mScarlet (green) and stained for F-actin (magenta). Scale bar: 10 μm. The numerical data underlying this figure can be found in **S1 Data**.
(TIF)

**S10 Fig. F/G actin ratio unaltered upon tagging β-actin with ALFA tag at position T229/ A230.** (**A**) Representative western blot of F-actin and G-actin fraction in parental HT1080 and

homozygous ALFA-β-actin HT1080 cells with and without ALFA-Nb-GFP. Total actin was used as a loading control. (**B**) Quantification of the F/G-actin ratio for β-actin from the western blots shown in **A**. Ratios were normalized against HT1080 parentals. Individual data points represent 3 independent western blots. Bars represent the mean value and error bars represent standard error of mean (SEM). The numerical data underlying this figure can be found in **S1 Data**.
(TIF)

**S11 Fig. F-actin co-sedimentation of myosin IIA and α-actinin.** (**A**) Representative western blot of myosin IIA in the F-actin and G-actin fraction in parental HT1080 and homozygous ALFA-β-actin HT1080. Total actin (tot-actin) was used as a loading control. (**B**) Representative western blot of α-actinin in the F-actin and G-actin fraction in parental HT1080 and homozygous ALFA-β-actin HT1080 cells. Total actin was used as a loading control.
(TIF)

**S12 Fig. Cell proliferation and migration are not affected by co-expressing the ALFA tag nanobody in ALFA-β-actin cells.** (**A**) Quantification of an MTT proliferation assay performed on ALFA-β-actin HT1080 cells with and without expressing the ALFA tag nanobody. Individual data points represent the average for 3 independent experiments. Bars represent the mean value and error bars represent standard error of mean (SEM). (**B**) Quantification of the wound closure assay shown in **C** indicating the distance closed in μm over time in ALFA-β-actin HT1080 cells and ALFA-β-actin HT1080 cells expressing ALFA-Nb-GFP. Large data points represent the mean of 3 independent experiments and the small data points represent the quantification of the individual images. Ten images per condition were acquired per experiment. (**C**) Representative widefield images of ALFA-β-actin HT1080 cells with and without expressing the ALFA tag nanobody at time point 0 h, 2 h, 4 h, 6 h, and 8 h after scratch induction. Scale bar: 30 μm. The numerical data underlying this figure can be found in **S1 Data**.
(TIF)

**S13 Fig. Position T229/A230 is a permissive site for epitope integration in all 6 human actin isoforms.** (**A**) Representative widefield immunofluorescence images of HT1080 with F-actin (magenta) and ALFA tag (green) in HT1080 cells that overexpress the ALFA tag in position T229/A230 in all 6 human actin isoforms. Scale bar: 15 μm. Scale bar zoom: 5 μm. (**B**) Representative widefield immunofluorescence images of RPE1 with F-actin (magenta) and ALFA tag (green) in HT1080 cells that overexpress the ALFA tag in position T229/A230 in all 6 human actin isoforms. Scale bar: 15 μm. Scale bar zoom: 5 μm.
(TIF)

**S14 Fig. IntAct β- and γ-actin recapitulate differential distribution of actin isoforms.** Representative Airyscan images of cells seeded on standard coverslips. Shown are parental HT1080 cells stained for β-actin (green) and γ-actin (magenta) and ALFA-β-actin and ALFA-γ-actin cells stained for ALFA tag (green) and F-actin (magenta). The zoom images present stress fibers, lamellipodia, and filopodia and the 32-color ratio images indicate the ratio between β- and γ-actin (parental cells) or the ratio between β- or γ-actin and total actin. Scale bar: 10 μm. Scale bar zoom images: 0.5 μm.
(TIF)

**S15 Fig. Sequence alignment of human β- and γ-actin, fission yeast actin (*S. pombe*) and budding yeast actin (*S. cerevisiae*).** Red residues indicate complete consensus among variants, blue indicates 1 substitution with the substitution indicated in black, and black columns indicates 2 substitutions.
(TIF)

**S16 Fig. Characterization of ALFA-tagged actins from yeast (*Saccharomyces cerevisiae* and *Schizosaccharomyces pombe*) and human (β- and γ-actin).** (**A**) Schematic overview of budding yeast (*S. cerevisiae*, top) and fission yeast (*S. pombe*, bottom) showing an extra copy of IntAct actins (integrated at auxotrophic marker locus or present in plasmid) in addition to the native *yeast actin* at its native locus. (**B**) Representative western blot showing expression of endogenous *S. cerevisiae* actin (ACT1) and β-IntAct, γ-IntAct, Sc-IntAct, Sp-IntAct expressed from either an integrating plasmid (pRS306) or centromeric plasmid (pRS316).
(TIF)

**S17 Fig. Budding yeast expressing only ALFA-Sc-actin as a sole actin copy is not viable, but co-expression of IntAct proteins result in similar growth rates.** (**A**) Spot assay image showing strains co-expressing budding yeast native *act1* with ALFA-Sc-actin in a low-copy and high-copy plasmid expression (2μ) (left) and spot assay image showing strains expressing ALFA-Sc-actin in a low-copy and high-copy plasmid expression (2μ) in the absence of native actin which was shuffled out using 5′-FOA media (right). (**B**) Spot assay image of strains co-expressing IntAct proteins with respect to wild-type strain.
(TIF)

**S18 Fig. IntAct actins localize to actin cortical patches in ALFA-Nb-mNeonGreen expressing yeast (*S. cerevisiae*).** Representative confocal images of time-lapse imaging of ALFA-Nb-mNeonGreen expressing budding yeast cells co-expressing β-IntAct, γ-IntAct, Sc-IntAct, and Sp-IntAct. Yellow dashed line indicates the outline of the yeast cell. Scale bar: 3 μm.
(TIF)

**S19 Fig. IntAct actins localize to actin cortical patches in ALFA-Nb-mNeonGreen expressing budding yeast.** (**A**) Representative confocal images of ALFA-Nb-mNeonGreen (green) expressing budding yeast cells co-expressing Sc-IntAct and Sp-IntAct. Stained for F-actin (magenta). Scale bar: 3 μm. (**B**) Representative montages of time-lapse imaging of ALFA-Nb-mNeonGreen expressing budding yeast cells co-expressing: β-IntAct and γ-IntAct treated by either DMSO or CK666 (200 μM). Yellow dashed line indicates the outline of the yeast cell. Scale bar: 3 μm.
(TIF)

**S20 Fig. IntAct actins localize to dynamic foci-like spots at cell poles and cell equator in ALFA-Nb-mNeonGreen expressing yeast (*S. pombe*).** Representative confocal images of time-lapse imaging of ALFA-Nb-mNeonGreen expressing fission yeast cells co-expressing β-IntAct, γ-IntAct, Sc-IntAct, and Sp-IntAct. Yellow dashed line indicates the outline of the yeast cell. Scale bar: 3 μm.
(TIF)

**S21 Fig. Actin patch lifetime does not change significantly upon tag integration in budding yeast.** (**A**) Representative time-lapse montages of budding yeast cells constitutively co-expressing ALFA-Sc-actin and ALFA-Nb-mNG. Native Abp1-3xmCherry was used as an actin patch marker to assess colocalization. Scale Bar: 3 μm. (**B**) Quantification of actin patch lifetimes across the indicated strains. For each strain, more than 15 actin patches were measured. Statistical analysis was performed using a one-way ANOVA post hoc Tukey's multiple comparison test. Box plots indicate median (middle line), 25th, 75th percentile (box) and minimum and maximum (whiskers). The numerical data underlying this figure can be found in **S1 Data**.
(TIF)

**S22 Fig. IntAct actins only localize to actin cortical patches in ALFA-Nb-mNeonGreen expressing yeast (*S. cerevisiae*) and not to actin cables.** Representative time-lapse montages

of budding yeast cells with constitutive expression Sc-IntAct and inducible expression of ALFA-Nb-mNeonGreen under GALS promoter. Galactose was added to a final concentration of 2% before start of imaging. Scale bar: 3 μm.
(TIF)

**S23 Fig. IntAct actins localize to cortical actin patches in wild-type budding yeast (*S. cerevisiae*) and wild-type fission yeast (*S. pombe*).** (A) Representative montages of budding yeast cells (*S. cerevisiae*) expressing β-IntAct and γ-IntAct. Cells were fixed and stained for F-actin (magenta) and ALFA tag nanobody (green). Scale bar: 3 μm. Scale bar zoom: 0.5 μm. (E) Representative montages of fission yeast cells (*S. Pombe*) expressing β-IntAct and γ-IntAct. Cells were fixed and stained for F-actin (magenta) and ALFA tag nanobody (green). Scale bar: 3 μm. Scale bar zoom: 0.5 μm.
(TIF)

**S1 Table. List of yeast strains used in this study.**
(TIF)

**S2 Table. Forward and reverse primers used for the PCR fragments of the FLAG actin overexpression constructs.**
(TIF)

**S3 Table. gRNAs and HDR templates used for insertion of FLAG-, AU5-, AU1-, and ALFA-tags in position 229–230 of β-actin, the ALFA tag in γ-actin and HDR template for ouabain resistance.** All oligos were inserted into the px330-hSpCas9 vector (Addgene, 42230) and the vector with the gRNA targeting *ATP1A1* was purchased (Addgene, 86611).
(TIF)

**S4 Table. List of plasmids constructed and used in this study.**
(TIF)

**S1 Movie. ALFA-tag colocalization with actin in HT1080 ALFA-β-actin cells transfected with LifeAct-GFP and ALFA-Nb-mScarlet.** Airyscan images were taken on a Zeiss LSM880 with 1 image per 15 s. Playback speed: 15 fps. Scale bar: 10 μm.
(AVI)

**S2 Movie. Actin flow at the lamellipodia in HT1080 parental cells transfected with LifeAct-GFP.** Airyscan images were taken on a Zeiss LSM880 with 1 image per 5 s. Playback speed: 15 fps. Scale bar: 4 μm.
(AVI)

**S3 Movie. Actin flow at the lamellipodia in HT1080 ALFA-β-actin cells transfected with LifeAct-GFP.** Airyscan images were taken on a Zeiss LSM880 with 1 image per 5 s. Playback speed: 15 fps. Scale bar: 4 μm.
(AVI)

**S4 Movie. Actin flow at the lamellipodia in HT1080 ALFA-β-actin cells transfected with ALFA-Nb-GFP.** Airyscan images were taken on a Zeiss LSM880 with 1 image per 5 s. Playback speed: 15 fps. Scale bar: 4 μm.
(AVI)

**S1 Data. Excel spreadsheet with individual numerical data underlying plots and statistical analyses.** The data are organized into separate sheets corresponding to the following figure panels: 1C, 1G, 2B, 2D, 2F, 2H, 4C, 4D, 4F, 5B, 5C, S3B, S5C, S5E, S7B, S8B, S10B, S12A, S12B,

and S21B.
(XLSX)

**S1 Raw Images. Uncropped, raw western blots underlying the western blots in the main and supplementary figures.** The data are organized into separate pages corresponding to the following figure panels: 2A, 2C, 2G, 3A, 3B, 3C, 3D, S4C, S5B, S5D, S5F, S6A, S6B, S10A, S11A, S11B, and S16B.
(PDF)

## Acknowledgments

We are indebted to NanoTag Biotechnologies GmbH, Göttingen, Germany for providing us with the ALFA nanobody expression construct. We also thank Prof. Jessica Henty-Ridilla and Prof. David Amberg (Upstate Medical University, New York, USA) for generously sharing yeast strains for our study. The authors further thank the Radboud Technology Center Microscopy of the Radboudumc and Bio-Imaging Divisional Facility at IISc Bengaluru for the use of their microscopy facilities. Also, we would like to thank Jessica Mazalo from the Erasmus MC for technical assistance with micropatterning experiments. A.D. acknowledges GATE Fellowship by IISc.

## Author Contributions

**Conceptualization:** Koen van den Dries.

**Data curation:** Maxime C. van Zwam, Anubhav Dhar, Willem Bosman, Wendy van Straaten, Suzanne Weijers, Emiel Seta, Ben Joosten, Saravanan Palani, Koen van den Dries.

**Formal analysis:** Maxime C. van Zwam, Anubhav Dhar, Saravanan Palani, Koen van den Dries.

**Funding acquisition:** Saravanan Palani, Koen van den Dries.

**Investigation:** Maxime C. van Zwam, Anubhav Dhar, Willem Bosman, Wendy van Straaten, Suzanne Weijers, Emiel Seta, Ben Joosten, Saravanan Palani, Koen van den Dries.

**Methodology:** Maxime C. van Zwam, Anubhav Dhar, Jeffrey van Haren, Saravanan Palani, Koen van den Dries.

**Project administration:** Saravanan Palani, Koen van den Dries.

**Supervision:** Saravanan Palani, Koen van den Dries.

**Validation:** Koen van den Dries.

**Visualization:** Koen van den Dries.

**Writing – original draft:** Maxime C. van Zwam, Anubhav Dhar, Saravanan Palani, Koen van den Dries.

**Writing – review & editing:** Maxime C. van Zwam, Anubhav Dhar, Saravanan Palani, Koen van den Dries.

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
