## [Editor Report · Decision Letter 0]

19 Sep 2023

Dear Dr van den Dries, 

Thank you for submitting your manuscript from Review Commons entitled "IntAct: a non-disruptive internal tagging strategy to study actin isoform organization and function" for consideration as a Methods and Resources Article by PLOS Biology. Please accept my apologies for the delay in getting back to you as we consulted with an academic editor about your submission. 

Your manuscript has now been evaluated by the PLOS Biology editorial staff, as well as by an academic editor with relevant expertise, and I am writing to let you know that we would like to invite you to submit a revised version in response to the reports at Review Commons.

However, before we can invite a revision, we need you to complete your submission by providing the metadata that is required for full assessment. To this end, please login to Editorial Manager where you will find the paper in the 'Submissions Needing Revisions' folder on your homepage. Please click 'Revise Submission' from the Action Links and complete all additional questions in the submission questionnaire.

To provide the metadata for your submission, please Login to Editorial Manager (https://www.editorialmanager.com/pbiology) within two working days, i.e. by Sep 21 2023 11:59PM.

Kind regards,

Richard

Richard Hodge, PhD

rhodge@plos.org

PLOS

---

## [Editor Report · Decision Letter 1]

21 Sep 2023

Dear Dr van den Dries,

Thank you very much for submitting your manuscript "IntAct: a non-disruptive internal tagging strategy to study actin isoform organization and function" for consideration as a Methods and Resources Article at PLOS Biology. As you know, your manuscript and plan of revision have been evaluated by the PLOS Biology editors and by an Academic Editor with relevant expertise. 

Based on your responses to the reviews from Reviews Commons, we would welcome re-submission of a revised version that takes into account the reviewers' comments according to your revision plan. 

Given the extent of revision needed, we cannot make any decision about publication until we have seen the revised manuscript and your response to the reviewers' comments at Review Commons. Your revised manuscript is also likely to be sent for further evaluation by the original reviewers.

We expect to receive your revised manuscript within 3 months. Please email us (plosbiology@plos.org) if you have any questions or concerns, or would like to request an extension. At this stage, your manuscript remains formally under active consideration at our journal; please notify us by email if you do not intend to submit a revision so that we may withdraw it.

**IMPORTANT - SUBMITTING YOUR REVISION**

*Re-submission Checklist*

*Published Peer Review*

*PLOS Data Policy*

*Blot and Gel Data Policy*

Sincerely,

Richard

Richard Hodge, PhD

rhodge@plos.org

REVIEWS:

Reviewer #1: In this study, the authors generate several variants of actin that are internally tagged with short peptide tags. They identify one particular position that is able to tolerate various tags of 5-10 amino acids and still shows largely unaltered behavior in cells. They study incorporation of their tagged actins into filaments, characterize the interactions of G-actin variants with different associated proteins and show that retrograde actin flow in lamellipodia and the wound healing response of epithelial cells is not affected by the tagged variants. They then apply the tagged actin to study subcellular distribution of different actin isoforms in mammalian and yeast cells. 

The identification of a specific site in the actin protein that tolerates variable peptide insertions is very exciting and of fundamental interest for all research fields that deal with cytoskeletal rearrangements and cellular morphogenesis. The result demonstrating the functionality of actin variants with peptides inserted between aa 229 and 230 are generally convincing and well done. In particular, the generation of CRISPR/Cas9 genome edited versions of beta- and gamma actin are impressive. I therefore generally support publication of this study. There are however several technical and conceptual issues that should be addressed to improve quality and scope of the study. I listed some specific comments below: 

**Major points**

- The biggest issue I have is the last section on the application of tagged actins to study isoform functions. In principle the application is very clear as there are simply no alternative ways to study isoform distribution in live cells. However, the experimental data are simply not convincing. What the authors define as "cortex" in Fig. 5A seems to rather represent cytosolic background mixed with radial fibers. I am not convinced that even the antibody staining with a relatively clear differential distribution of beta and gamma really shows a genuine accumulation of one isoform on stress fibers. It seems to me that the beta-actin staining has as higher cytosolic background and is generally weaker (gamma nicely labels transverse arcs), which reduces signal/noise and therefore yields a relatively increased level in areas with less-bundled actin. My suggestion is to select more clearly defined actin structures and to use micro-patterned cells to normalize the otherwise obstructing variability in actin organization. Possible structures would be cortical arcs in bow-shaped cells, lamellipodial edges (HT1080 seem to make very nice and large lamellipodia) or cell-cell contacts (confluent monolayer, provided cells don´t grow on top of each other). Stress fibers are possible but need to be segmented very precisely and I did not see any details on this in the methods section. For Fig. 5D: I assume cells were used where only one isoform was tagged? This is technical weak and the double-normalization is probably blurring any difference that might be occurring. Why not use a double-tagging strategy with ALFA/FLAG or ALFA/AU5 tags to exploit the constructs introduced in the previous figures? Also, the unique selling point of the strategy is the possibility of actual live imaging of specific isoforms. Cells that have stably integrated double tags and then transiently express nanobodies for ALFA and either AU5 or FLAG (or other if those don't exist) would make this possible. Considering the work already done in this manuscript, such an approach should actually be possible - did the authors attempt this or is there are reason it is not discussed? If double tagged cells are not possible for some reason it should at the very least be possible to combine ALFA-detection with the specific antibody against the other isoform and get rid of the double normalization. 

- The authors make a point of comparing the internally tagged actin to N-terminal tags that are mostly functional but have been shown to affect translational efficiency. I would strongly suggest to include N-terminally tagged actin as control for all assays in this study. Also for the physiological assays (retrograde flow, wound healing), a positive control is missing that shows some effect. Previous studies showed defects with transiently expressed actin with an N-terminal GFP. As retrograde flow measurements are very sensitive to the exact position of the kymographs and wound healing assays is a very crude and indirect readout, such a positive control is essential. 

- Expression of tagged actins in yeast is a very nice idea but it would be far more informative to express the tagged forms as the only copy of actin. This can either be done by directly replacing endogenous actin gene in S. cerevisiae, or (if the tagged versions are not viable) - using the established plasmid shuffle system (express actin on counter-selectable plasmid, then knock out endogenous copy and introduce additional plasmid with tagged actin, then force original plasmid out). In the presence of endogenous S. cerevisiae actin the shown effects are very hard to interpret as nothing is known about relative protein levels (endogenous vs. introduced). Also, if constitutive expression of the ALFA nanobody is harmful for integration into cables, why not perform inducible expression of the nanobody and observe labeling after induction. For the live imaging a robust cable marker is needed, like Abp140-GFP. Finally, indicate the sequence differences between the used actin forms in yeast (supplementary figure with sequence alignment and clear indication of all variations) 

- As the authors clearly show good integration of several tagged actins into filaments I would expand the structural characterization: perform alpha fold predictions of actin monomer structures including the various tags to show the expected orientation. It is striking that the only integration site that seems to work well is at the last position of a short helix, indicating that the orientation of the integrated peptide might be fixed in space and be optimal to minimize interference. Also, a docking of the tag onto the recently published cryoEM structures of the actin filament should be shown to indicate where it resides compared to tropomyosin or the major groove where most side binding proteins seem to bind.

- For any claims regarding usability of tagged variants for isoform research it would be very important to characterize the known posttranslational modifications of tagged actin variants - are the differences between beta and gamma maintained on this level as well? 

**Technical issues** 

- There is no scale for the color coding in Fig. 5A, B 

- The y-scales for Fig. 5C and D need to be identical to allow direct comparison 

- Pearson coefficient should not be normalized to a control value as its already a dimensionless parameter. Always report actual R-value - also remove R2 values for Pearson as this makes no sense in this context (not sure if it was a typo or intended). 

- All values on subcellular regions (like stress fiber or cortex) dependet critically on the way thesese regions were thresholded or identified. Provide all details on how this was done in the methods section and ensure that adequate background subtraction and normalization is applied. Optimally, an unbiased (AI or automated) approach based on simple image statistics is used for this to avoid personal bias. 

- In Fig. 2A only heterozygous FLAG-actin cells are used. Why not use a homozygous line (for both beta and gamma actin)? The nice band shift of the FLAG version would allow the precise quantification of the fraction of total actin covered by beta and gamma actin, which then could provide some additional info for the apparently weaker beta staining in Fig. 5 (if beta expression is simply weaker). This would be a very simple and useful advantage of the internal tags that could be widely applied. 

- Fig. 3: control with N-terminal tag is missing. Also, why is it not possible to assay filament binding factors like Myosin, Filamin or alpha actinin - instead of co-IP a simple cosedimentation assay with cell extracts in F-buffer should pick up any major difference in decoration of filaments containing the ALFA tag. Using two speeds for centrifugation it might even be possible to observe effects on filament bundling. The best approach for this would of course be to purify tagged actins and perform in vitro assays but this is clearly beyond the scope of what the authors intended here. I personally think that a broad acceptance of the marker will only come once the biochemistry has been sufficiently characterized so this is a future direction I would strongly encourage. 

- Fig. 2A has no loading control – 

- The RPE-1 data are confusing as several constructs show very different localization (completely cytosolic) to HT1080 cells and there is no possible explanation given for this. Maybe simply remove this data set? 

- The angel measurements for lamellipodial actin is not very meaningful: the angel is determined for the radial bundles, which do not correspond to the Arp2/3 angel of single filaments and is likely the results of different nucleation factors, I would suggest to remove this. If angel measurement are really intended, cryoEM needs to be performed. 

- Replace all SEM with SD values - use at least 3 biological replicates (4D SEM of n=2) 

**Minor points** 

- Intro: after listing all the details already understood on actin isoforms it is not very convincing to simply state the molecular principles remain largely unclear (l 34) - maybe better "there is no way to study actin dynamics due to current limitations of specific antibodies to fixed samples. Interesting option would be actually to develop nanobodies that are isoform specific �

 - L 71: "involved" in the kinetics is not a good term - maybe affects or regulates.... 

- L148: "suspect" instead of "expect" - this clonal variation is actually a big danger of the employed approach as possible defects in actin organization could be masked by compensatory changes - it would generally be good to show critical data for at least 3 independent clones to rule out dominant selection effects. 

**Referees cross-commenting** I completely agree with the comments by reviewer 2 on the various missing controls - adding several or all of those will make the results much more convincing. The key for the adaptation of any new actin probe will be the level of confidence researchers have on the documented effects. Even some negative effects on actin behavior (I am sure there will be some) should not prevent usage of the strategy as long as there is robust and convincing documentation of those effects. I also agree that including some basic in vitro characterization will go a long way to convince people dierectly working on actin (there is a very high level of biochemical understanding in that field). 

Significance:* Very useful finding that can be applied to any question related to actin dependent cellular processes 

(morphogenesis, cell division, cell polarization, cell migration etc.) 

*Strength:* main finding convincing, strong genome edited cell lines 

*Limitations:* application to study of isoforms very limited and data not convincing, statistics and image quantifications need improvement 

*Advance:* identify new location for integral tagging of actin, which was not really possible before. The main relevance is for fundamental cell biology but the approach can also be applied to the study of disease variants in actin. 

*Audience:* general cell biology - very broad interest

Reviewer #2: Actin is highly sensitive to modifications, and tagging it with fluorescent proteins or even smaller motifs can affect its function. The most well-known example of this is that fission yeast where actin has been replaced with GFP-actin are inviable (Wu and Pollard, Science 2005) because the labeled actin cannot incorporate into the formin-dependent filaments that make up the cytokinetic ring. Subsequent experiments revealed that formins filter out GFPactin monomers, as well as monomers that are labeled with smaller fluorescent motifs (Chen et al, J. Structural Biology 2012). Further, attempts to make mammalian cells lines where GFP-beta-actin was knocked into one allele resulted in extreme down-regulation of the GFPlabeled actin, indicating that there is some implicit toxicity with the labeled version. To my knowledge, all attempts at making homozygous GFP-actin knock-ins have been unsuccessful. Therefore, while GFP-actin or other labeled variants can be over-expressed in many different cell types with some success, there is always the question of how faithful the labeled actin represents bona fide actin localization and dynamics. 

To address this van Zwam et al. have developed a clever strategy of screening actin for internal motifs that can tolerate incorporation of a tag without affecting its function. They appear to have found a good candidate, named IntAct, and provide evidence that this tagging position allows the actin to be functional in both human and yeast cells. The work is very promising, and many of the assays performed satisfy the criteria of rigor and reproducibility. Importantly, the authors have created knock-in human cell lines where the tagged actin is expressed at normal levels, including a double allele knock-in that is viable and has normal proliferation and motility. Additionally, the authors show that labeled S. cerevisiae actin can incorporate into actin cables, which are formin dependent. IntAct constructs were shown to interact with several well-known actin binding proteins and localized well to many different actin structures. There was also interesting data obtained from tagging both beta and gamma actin in human cells. However, as an actin scientist eager for new probes to visualize actin in cells, there are still questions about the functionality of these probes. Addressing these issues, listed below, would alleviate the concerns I still have about IntActs after going through the manuscript. IntActs have the potential to have a large impact on cytoskeletal research if it can be rigorously documented that they are functionally as close to unlabeled actin as possible.

**Concerns:** 

1. There are no negative controls performed for either the fixed or live-cell imaging of IntAct. Since the fixed cell data is heavily reliant on the presence of flag-labeled puncta at actin filaments, it is important to show that the immunocytochemistry protocol doesn't produce anything that would mimic the localization of actin. For the live cell data, there has been no effort made to show that the binding of the nanobody to the ALFA tag on InAct is specific. 

2. The homozygous ALFA-tagged IntAct cells have a 50% reduction in the amount of actin expression (Fig. 2D). What is the F:G ratio in these cells? The F:G measurement is only shown for the FLAG-tagged heterozygous IntAct cells, which have the worst co-localization with phalloidin (Fig. 2F) and were not used for subsequent figures. I appreciate that motility and proliferation were measured and shown to not be affected (Fig. 4D,E) , but in our lab reducing the amount of polymerized actin by 50% (which may be more in ALFA-tagged IntAct cells if the F:G changes) has catastrophic effects on other cytoskeletal and organelle systems. Since the homozygous ALFA IntAct cells are the main ones used in the manuscript, they should be the ones that are fully characterized. 

3. It is not addressed if expressing the ALFA-Nb-GFP construct in ALFA-IntAct cells alter actin properties? This is essential information for live cell imaging experiments. 

4. It is not addressed how much of the ALFA-IntAct gets labeled with ALFA-Nb-GFP and how uniform the labelling. 

5. To assess lamellapodia architecture, "branched actin angle" is measured using AiryScan imaging of actin filaments. This type of microscopy does not offer the ability to image individual actin filaments; what is actually being measured is the orientation of actin bundles to each other. It should be impossible to image the orientation of actin filaments in Arp2/3 dendritic networks and it is surprising that the measurements average to 70 degrees. A suitable substitute for this would be to measure the size and amount of F-actin in phalloidinstained lamellipodia using kymograph analysis. 

6. Was it possible to make an IntAct gene substitution in yeast? Also, while this is not necessary for this manuscript, making a fission yeast strain where actin has been substituted with IntAct and demonstrating that IntAct gets incorporated into the cytoplasmic ring and into Cdc12p-polymerized filaments would alleviate MANY potential concerns people would have about these probes by directly assessing situations were other labeled actins have been documented to fail. Along the same lines, it would have been nice to see a comparison in some of the assays of ALFA-IntAct and GFP-actin or another labeled actin variant.

Reviewer #3: **Summary:** This paper tackles a new strategy to tag actin in cells, by identifying that incorporation of a tag of moderate size in subdomain 4 of actin minimally affects actin dynamics in cells, and does not perturb its interaction with known partners, as observed in pull-down assays. 

**Major comments:** 

The paper is interesting and experiments are convincing. My main concerns are the following: 

- Varland et al, is reporting a phosphorylation on Thr229 : I think the authors should mention and discuss this potential PTM that could be affected in IntAct. 

- The sequence in subdomain 4 (the alpha helix containing T229A230) is extremely conserved in animals, as well as in between the 6 human actin isoforms. This usually indicates a strong selection pressure on the residues. I think the authors should discuss how surprising it is that the T229A230 position can accomodate various tags while it is probably the place of interaction with other proteins and is playing an important role in the mechanical structural integrity of the actin itself. 

- It is now well established that actin plays active and important roles in the nucleus : is ALFA-actin correctly translocated to the nucleus?

- OPTIONAL: one may regret that there is no classical in vitro assays, such as pyrene assays to assess some kinetcis parameters on epitope-tagged actins. I guess this would make the paper a bit too large. Although, it will prove useful to better understand how much formin activity is affected (see below) 

**Minor comments:** 

Below are points that could be addressed by the authors to improve the manuscript readability and highlight some important points that are sometimes missing or are not properly discussed : 

- line 40 "...but the distinct N-terminal epitope is not available under native conditions preventing" is a bit too obscure. Can the authors say clearly what is meant by 'native conditions' ? 

- figure 1A : make a clearer correspondance between the number shown in panel A and the amino acid numbers displayed in panel C and G. - figure 1A : it could be informative to indicate subdomains in this panel. 

- figure 1C : normalized correlation cell : I am not sure I understand how the normalization of the Pearson coefficient is done. It is therefore not clear how can it >1 or >-1 ? This should be clearly explained in the method section of the paper. 

- figure S4 : comes a bit too early when ALFA-actin has not been yet introduced in the main text. Please, reposition this part or provide data with the FLAG-tag version. - section starting line 121 : this section should be better motivated = Why are different tags being tested ? This comes later in the discussion, but the reader fails at following the reasoning/motivation here. 

- figure 2D, line 145 "We also evaluated actin protein expression in the homozygous ALFA-βactin cells and this showed that the total amount of β-actin was slightly lower in the ALFA-βactin cells compared to parental HT1080 cells (Fig. 2C-D)." 'Slightly' is not a very quantitative nor accurate term. please rephrase. Besides, a statistical test for the paired data would also be informative. Besides, data in figure S6B-D indeed show a correlated increase in the expression of Gamma-actin that compensate for the decrease in the Beta-actin level in ALFA-Beta-actin. Can the authors explain why they conclude otherwise ? 

- figure S7B: I am not ure anyone has ever reported measurement of angle of branched actin filament using epifluorescence microscopy. I would remove this panel, or the authors should explain how this measurement can be done objectively. 

- Figure 2F : can the authors comment on the (significant ?) lower value for FLAG-tag actin ? 

- line 205 "The results from these experimentsshow that both DIAPH1 and FMNL2 associate with ALFA-β-actin (Fig. 3D),". It is not so obvious that these formins directly interact with monomeric actin via their FH2 domains in co-immunoprecipitation assays. It might very well be mediated by the interaction with profilin, that in turn bind to the FH1 domain of formins. For me, this assay does not make a correct proof that epitope-labelled actin do not interfere with formin activity. 

- figure 5C&D : both graph should use the same scale for the y-axis for easier comparison. 

- figure 5D: I think the way the ratio is performed is misleading. Why not look at the Beta/Gamma ratio using the isoform specific antibodies used in parental cells, and show the results for ALFA-Beta-actin and for ALFA-Gamma-actin separately ? 

- The limitation observed for unbranched cables in yeast that nanobody-tagged ALFA-actin does not incorporate correctly should be discussed and stressed further in the discussion, as it might prove to be a strong limitation for live-cell imaging to reliably study any type of actin networks. 

*General assessment:* This paper provides a new tagging strategy to monitor actin activity in cells, by specifically inserting the tag along the amino acid sequence. 

*Advance:* This is a very useful tool, as most existing available probes bind to actin in regions that are common to many other actin binding proteins. The authors provide extensive experiments to validate that tagged-actin are functional and do not perturb the actin expression level, actin network architecture nor dynamics. 

*Audience:* This research paper will be of interest to a rather broad audience (many cell biologists) that are either sutyding actin dynamics or know that actin is involved in the cell functions they study. 

*Expertise:* My expertise is in vitro actin biochemistry.

---

## [Decision Letter · Decision Letter 2]

31 Jan 2024

Dear Dr van den Dries,

Thank you for your patience while we considered your revised manuscript "IntAct: a non-disruptive internal tagging strategy to study actin isoform organization and function" for publication as a Methods and Resources Article at PLOS Biology. Please accept my apologies for the delays that you have experienced during the peer review process due to the Christmas break last month. This revised version of your manuscript has been evaluated by the PLOS Biology editors, the Academic Editor and the original reviewers at Review Commons.

Based on the reviews, I am pleased to say that we are likely to accept this manuscript for publication, provided you satisfactorily address the following data and other policy-related requests that I have provided below (A-H):

(A) We would like to suggest the following modification to the title: 

“IntAct: a non-disruptive internal tagging strategy to study the organization and function of actin isoforms”

(B Please ensure that the figure legends specify how many biological replicates were performed for each experiment, particularly for the representative Western Blot images. 

(C) Please also ensure that each of the relevant figure legends in your manuscript include information on *WHERE THE UNDERLYING DATA CAN BE FOUND*, and ensure your supplemental data file/s has a legend.

(D) We require the original, uncropped and minimally adjusted images supporting all blot and gel results reported in the following Figures:

Fig 2A, 2C, 2G, 3A-D, S4C-D, S5B, S5D, S5F, S6A-B, S10A, S11A-B, S16B

We will require these files before a manuscript can be accepted so please prepare and upload them now. Please carefully read our guidelines for how to prepare and upload this data: https://journals.plos.org/plosbiology/s/figures#loc-blot-and-gel-reporting-requirements

(E) For figures containing FACS data (Fig. S4A-B), please provide the FCS files and a picture showing the successive plots and gates that were applied to the FCS files to generate the figure. We ask that you please deposit this data in the FlowRepository (https://flowrepository.org/) and provide the accession number/URL of the deposition in the Data Availability Statement in the online submission form.

(F) Please ensure that your Data Statement in the submission system accurately describes where your data can be found and is in final format, as it will be published as written there. This includes referencing where the flow cytometry data in the FlowRepository and data file provided in the Supplementary Information.

(G) We ask that the following sentence in the manuscript file is removed, since all of the primary data should be provided with the submission:

“All primary data supporting the conclusions made are available from the authors on request.”

(H) Please note that per journal policy, the model system/species studied should be clearly stated in the abstract of your manuscript. 

We expect to receive your revised manuscript within two weeks. 

*Published Peer Review History*

*Press*

Sincerely,

Richard

Richard Hodge, PhD

rhodge@plos.org

Reviewer remarks:

Reviewer #1: The authors have done a very thorough job to address all my concerns and I now fully support publication of this exciting new tool.

Reviewer #2: The author's have throughly responded to the reviewers' comments and the revised manuscript has my endorsement for publication in PLoS Biology. It is unfortunate that it was not possible to make yeast strains with IntAct as the sole actin, that would have been the evidence needed that IntActs will be revolutionary for the study of the cytoskeleton. Still, this work makes a sufficient advance forward in the visualization of actin that it will still be highly exciting to cytoskeletal biologists, particularly those working with mammalian cells. The work is thorough and well-documented, with numerous experiments done to validate IntActs and demonstrate their potential utility. Good job on the revision!

Reviewer #3: I am very satisfied with the revised version of the manuscript. I hope this tagged version of actin will be tested extensively by other labs around the world to validate its use in other cell types or organisms.

---

## [Editor Report · Decision Letter 3]

16 Feb 2024

Dear Dr van den Dries,

On behalf of my colleagues and the Academic Editor, Carole Parent, I am pleased to say that we can in principle accept your manuscript for publication, provided you address any remaining formatting and reporting issues. These will be detailed in an email you should receive within 2-3 business days from our colleagues in the journal operations team; no action is required from you until then. Please note that we will not be able to formally accept your manuscript and schedule it for publication until you have completed any requested changes.

PRESS

Best wishes, 

Richard Hodge, PhD

rhodge@plos.org

PLOS
